# A Language-Agent Approach to Formal Theorem-Proving

## Abstract

Language agents, which use a large language model (LLM) capable of in-context learning to interact with an external environment, have emerged as a promising approach to control tasks. We present a language-agent approach that offers state-of-the-art performance in formal theorem-proving. Our method, Copra, uses a high-capacity, black-box LLM (GPT-4) as part of a policy for a stateful backtracking search. During the search, the policy can select proof tactics and retrieve lemmas and definitions from an external database. Each selected tactic is executed in the underlying proof framework, and the execution feedback is used to build the prompt for the next policy invocation. The search also tracks selected information from its history and uses it to reduce hallucinations and unnecessary LLM queries.

We evaluate Copra on the `miniF2F` benchmark for Lean and a set of Coq tasks from the Compcert project. On these benchmarks, Copra is significantly better than one-shot invocations of GPT-4, as well as state-of-the-art models fine-tuned on proof data, at finding correct proofs quickly.

## 1 Introduction

Automatically proving formal theorems (Newell et al., 1957) is a longstanding challenge in computer science. Autoregressive language models (Polu & Sutskever, 2020; Han et al., 2021; Yang et al., 2023) have recently emerged as an effective approach to this problem. Such models are trained on proofs written in frameworks like Coq (Huet et al., 1997) or Lean (de Moura et al., 2015), which allows proof goals to be iteratively simplified using a set of *tactics*. Theorem-proving then amounts to generating a sequence of tactics that iteratively "discharges" a given proof goal.

A weakness of this method is that it does not model the *interaction* between the model and the underlying proof framework. The application of a tactic is an *action* that changes the state of the proof and the interpretation of future tactics. By ignoring these game-like dynamics, autoregressive models miss out on a valuable source of feedback and end up being more susceptible to hallucinations.

In this paper, we show that the nascent paradigm of *large-language-model (LLM) agents* (Yao et al., 2022; Wang et al., 2023; Shinn et al., 2023) can help address this weakness. Here, one uses an LLM as a *agent* that interacts with an external environment. Information gathered through interaction is used to update the LLM's prompt, eliciting new agent behavior because of in-context learning.

Our approach, called Copra[1] (Figure 1), uses an off-the-shelf, high-capacity LLM (GPT-4 (OpenAI, 2023a)) as part of a policy in that interacts with a proof environment like Coq or Lean. At each time step, the policy consumes a textual prompt and chooses to use an available tactic, or backtrack, or retrieve relevant lemmas and definitions from an external corpus. When the policy selects a tactic, we "execute" it using the underlying proof assistant. The feedback from the execution is used to construct a new prompt for the policy, and the process repeats.

Copra goes beyond prior language-agent methods in using domain knowledge and information from the search history to use LLM queries frugally. When tactics fail, the policy records this information and uses it to avoid future failures. The policy also has access to a symbolic procedure that checks if one goal is "simpler" than another. A tactic is only used when it simplifies the agent's proof obligations (ruling out, among other things, cyclic tactic sequences).

---

[1] Copra is an acronym for "In-**co**ntext **Pr**over **A**gent".

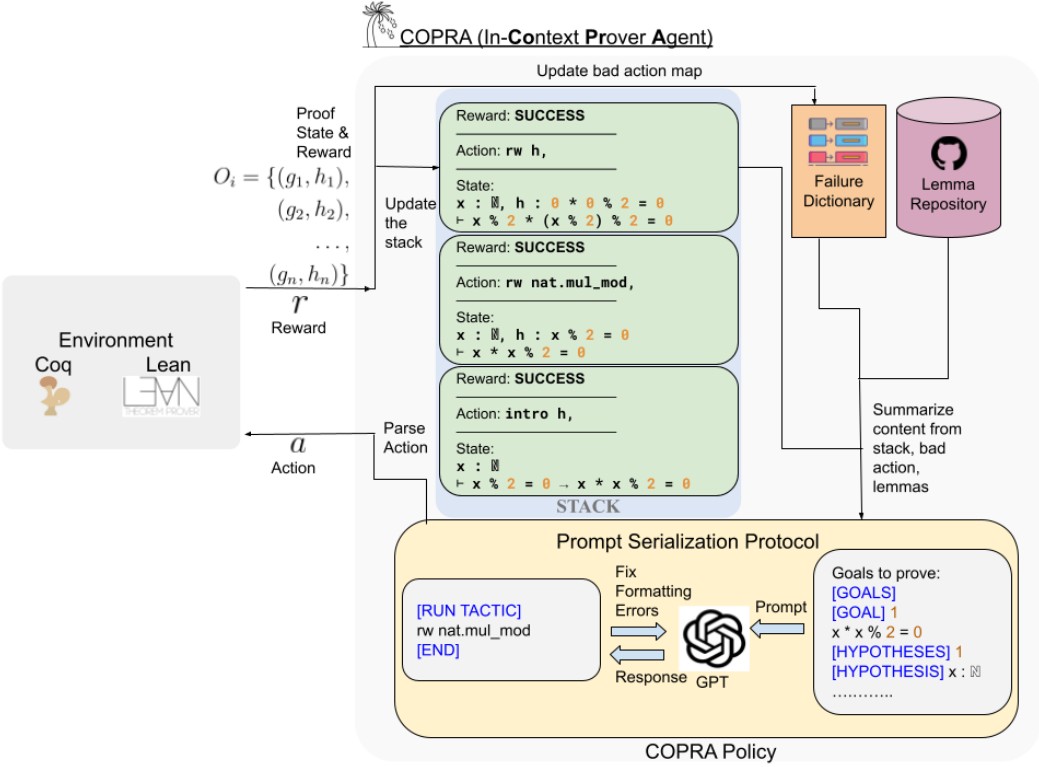

Figure 1: An overview of COPRA. The system implements a policy that interacts with a proof environment (Coq or Lean). Internally, a COPRA policy consists of an LLM (GPT-4), a stack-based backtracking search, a retrieval mechanism, a dictionary tracking past failures, and a prompt serialization protocol that constructs LLM prompts using the stack and environment feedback and parse LLM outputs into actions.

We have integrated COPRA with both the Coq and the Lean environments. We evaluate the system using the `miniF2F` (Zheng et al., 2021) benchmark for competition-level mathematical reasoning in Lean and a set of Coq proof tasks (Sanchez-Stern et al., 2020) from the Compcert (Leroy, 2009) project on verified compilation. Using a new metric called *prove-at-k-guidance-steps*, we show that COPRA can converge to correct proofs faster than competing approaches, including the state-of-the-art models (Yang et al., 2023; Sanchez-Stern et al., 2020) trained on formal proof data. We also show that when COPRA fails, it fails quicker than the baseline methods.

To summarize our contributions, we offer: (i) The first approach to formal theorem-proving that leverages LLMs while also modeling interactions between the model and the underlying proof framework; (ii) the first language agent, from any domain, to integrate LLM policies with a search that minimizes LLM queries and hallucinations by tracking domain-specific information from the past; and (iii) an implementation of COPRA that interacts with the Coq and Lean proof environments, and an evaluation on two domains — mathematics competition problems and formal verification — that shows COPRA to find proofs faster than competing approaches.

## 2 THEOREM-PROVING AS A CONTROL PROBLEM

### 2.1 BACKGROUND ON THEOREM-PROVING

A *formal proof* starts with a set of unmet *obligations* stated in a formal language and applies a sequence of *proof tactics* to progressively eliminate these obligations. Each obligation $o$ consists of a *goal* $g$ and a *hypothesis* $h$. The goal $g$ consists of the propositions that need to be proved in order to meet $o$; the hypothesis $h$ captures assumptions that can be made in the proof of $g$. The prover's long-term objective is to reduce the obligations to the empty set.

We illustrate this process with the example in Figure 2-(a). This example shows a Lean (de Moura et al., 2015) proof, automatically generated using COPRA, of a basic theorem about modular arithmetic. The proof first applies the `intro` tactic, which changes a goal $P \rightarrow Q$ to a hypothesis $P$ and a goal $Q$. Next, it applies the `rw` (rewrite) tactic, which gives a way to apply substitutions to goals and hypotheses, several times. It ends with the application of the `refl` (reflexivity) tactic, which eliminates goals that say that a value is equal to itself.

Existing LLM-based approaches to automatic theorem-proving view such proofs as purely syntactic artifacts. However, the rigorous semantics of proofs can be difficult to learn using such an approach, leading to the generation of incorrect proofs. Figure 2-(c) shows a GPT-4-generated incorrect proof of our theorem.

(a)
```
theorem mod_arith_2
(x : ℕ) : x % 2 = 0
    → (x * x) % 2 = 0
:=
begin
  intro h,
  rw nat.mul_mod,
  rw h,
  rw nat.zero_mul,
  refl,
end
```

(b)
```
x: ℕ
h: x % 2 = 0
⊢ x * x % 2 = 0
```

(c)
```
begin
  intro h,
  have h1 : x = 2 * (x
      / 2)
    := (nat.
    mul_div_cancel' h)
    .symm,
  rw h1,
  rw nat.mul_div_assoc
    _
    (show 2 | 2, from
    dvd_refl _),
  rw [mul_assoc, nat.
    mul_mod_right],
end
```

Figure 2: (a) A Lean theorem and a correct proof found by COPRA. (b) Proof state after the first tactic. (c) An incorrect proof generated by GPT-4.

## 2.2 A MARKOV DECISION PROCESS FORMULATION

By contrast, COPRA is based on a view of automatic theorem-proving as a *control problem*. Like prior work on reinforcement learning (RL) for proof synthesis (Wu et al., 2021), we view a theorem-prover as a *policy* that interacts with a stateful proof environment (e.g., Lean) and model the interaction between the policy and the environment as a deterministic Markov Decision Process (MDP). We depart from prior RL-based work for theorem-proving by imposing a partial order on MDP states, adding *execution feedback* in error states, and allowing history-dependent policies.

Now we describe the different components of our *proof MDP*.

**States.** As before, let an *obligation* be a pair $(g, h)$, where $g$ is a goal and $h$ a hypothesis. A *state* of the MDP is either a special symbol called *error* or a set $O = \{o_1, \ldots, o_k\}$ of obligations $o_i$. The MDP has a unique *initial state* $o_{in}$ with a single obligation $(g_{in}, h_{in})$, where the goal $g_{in}$ and the hypothesis $h_{in}$ are extracted from the user-provided theorem that we are trying to prove. Its unique *final state* `QED` is the empty obligation set. The special *error* symbol is accompanied by textual feedback in the form of an execution error message, *execution feedback*, from the proof environment.

Following Sanchez-Stern et al. (2020), we define a partial order $\sqsubseteq$ over states that defines when a state is "at least as hard" than another and use it to avoid actions that do not lead to progress in the proof. Formally, for states $O_1$ and $O_2$ with $O_1 \neq error$ and $O_2 \neq error$, $O_1 \sqsubseteq O_2$ iff

$$\forall \, o_i = (g_i, h_i) \in O_1. \, \exists o_k = (g_k, h_k) \in O_2. \, g_k = g_i \wedge (h_k \rightarrow h_i).$$

Intuitively, $O_1 \sqsubseteq O_2$ if for every obligation in $O_1$, there is a stronger obligation in $O_2$. We assume we have an efficient symbolic procedure that can check this relationship for any pair of states. The procedure is *sound*, meaning that if it reports $O_1 \sqsubseteq O_2$, the relationship actually holds. However, it is *incomplete*, i.e., it may not detect all relationships of the form $O_1 \sqsubseteq O_2$.

**Actions and Transitions.** The actions in our MDP are the proof environment's *tactics*. The transition function $T(O, a)$ determines the result of applying an action $a$ to a state $O$. When $a$ is a tactic, we assume the underlying proof environment to return a state $O'$ that results from applying $a$ to $O$. If $a$ is a "bad" tactic, then $O'$ equals *error*; otherwise, $O'$ is a new set of obligations. We assume that our agent can evaluate $T(O, a)$ for any state $O$ and action $a$. While this assumption is unacceptable in many MDP problems, it is reasonable in the theorem-proving setting.

**Rewards.** As usual, we assume a *reward function* $R(O, a)$ that evaluates an action $a$ at a state $O$. Concretely, we consider rewards of the form $R(O, a) = \tilde{r}$, where $\tilde{r}$ is a very high positive value if

$T(O, a) = \text{QED}$, a high negative value if $T(O, a) = error$, and a small negative value otherwise. The small negative reward on the successful execution of the action incentivises smaller proofs.

**Histories and Policies.** A *history* of length $N$ is a sequence

$$h = \langle (O_0, a_0, O'_0, r_0), (O_1, a_1, O'_1, r_1), \ldots, (O_{N-1}, a_{N-1}, O'_N, r_N) \rangle$$

such that $O_0 = O_{in}$ and for all $i$, $r_i = R(O_i, a_i)$ and $O'_i = T(O_i, a_i)$. Intuitively, a history records the interactions between the prover agent and the proof environment up to a point of time. We denote by $h_i$ the $i$-th prefix of $h$. For example, $h_0 = \langle \rangle$, $h_1 = \langle (O_0, a_0, O'_0, r_0) \rangle$, and so on.

A *policy* is a probabilistic function $\pi$ that maps histories to distributions over pairs $(O, a)$, where $O$ is a state and $a$ is an action. Intuitively, at each point, the policy determines the next query to make to the proof environment. A policy can have an internal state as well as access to external knowledge (specifically, a lemma database). A *trajectory* of a policy $\pi$ is a history $h$ as above such that for each $i$, $\mathbf{Pr}[\pi(h_i) = (O_i, a_i)] > 0$. Letting each $r_i = \tilde{r}_i$, the *reward* from a trajectory is simply the average $\frac{1}{N} \sum_i \tilde{r}_i$. We define the *aggregate reward* of $\pi$ as the expected reward from trajectories sampled from $\pi$.

**Language Agents.** Given our setup, one can naturally pose the problem of reinforcement-learning a policy with optimal aggregate reward. In this paper, we do not take on this problem. Instead, we consider a fixed policy — a wrapper around a pretrained LLM (GPT-4) that can learn in-context — and show that this policy can achieve a high reward. It is this policy that defines our *language agent*.

## 3 THE COPRA AGENT

A COPRA policy has access to an LLM (in practice, GPT-4) and performs a depth-first search. During the search, it records information about failed actions. It also uses the $\sqsubseteq$ relation over states to checks that it is making progress on the proof.

Figure 3 shows pseudocode for such a policy. The policy maintains a stack of MDP states and a "failure dictionary" $Bad$ that maps a state to a set of actions that are known to be "unproductive" at the state. At each search step, the algorithm pushes the current state on the stack and retrieves external lemmas and definitions relevant to the state. After this, it repeatedly serializes the stack and $Bad(O)$ into a prompt and feeds it to the LLM. The LLM's output is parsed into an action, and the agent executes it in the environment.

One outcome of the action could be that the agent arrives at QED. Alternatively, the new state could be an error or represent obligations that are at least as hard as what is currently on the stack (for example, this could be because of a cycle

```
COPRA(O)
1   PUSH(st, O)
2   ρ ← RETRIEVE(O)
3   for j ← 1 to t
4       do p ← PROMPTIFY(st, Bad(O), ρ, r)
5           a ∼ PARSEACTION(LLM(p))
6           O' ← T(O, a), r ← R(O, a)
7           if O' = QED
8               then terminate successfully
9               else if O' = error or
                        ∃O'' ∈ st. O'' ⊑ O'
10                      then add a to Bad(O)
11                      else COPRA(O')
12  POP(st)
```

Figure 3: The search procedure in COPRA. $T$ is the environment's transition function and $R$ is the reward function. $st$ is a stack, initialized to be empty. $Bad(O)$ is a set of actions, initialized to $\emptyset$, that are known to be bad at $O$. LLM is an LLM, PROMPTIFY generates a prompt, PARSEACTION parses the output of the LLM into an action (repeatedly querying the LLM in case there are formatting errors in its output), and RETRIEVE gathers relevant lemmas and definitions from an external source. The procedure is initially called with argument $O_{in}$.

in a tactic). In this case, the agent rejects the new state. Otherwise, it recursively continues the proof from the new state. After issuing a few queries to the LLM, the agent backtracks.

**Prompt Serialization Protocol.** The routines PROMPTIFY and PARSEACTION together constitute the *prompt serialization protocol* and are critical to the success of the policy. Now we elaborate on these procedures.

| | | | | |
|---|---|---|---|---|
| Agent Prompt (State) | Goals to prove: [GOALS] [GOAL] 1 x * x % 2 = 0 [HYPOTHESES] 1 [HYPOTHESIS] x : ℕ [HYPOTHESIS] h : x % 2 = 0 | | | Goals to prove: [GOALS] [GOAL] 1 x % 2 * (x % 2) % 2 = 0 [HYPOTHESES] 1 [HYPOTHESIS] x : ℕ [HYPOTHESIS] h : x % 2 = 0 |
| Agent Prompt (Stack) | [LAST STEP] intro h, | [STEPS] [STEP] intro h, [LAST STEP] rw h, | [STEPS] [STEP] intro h, [INCORRECT STEPS] [STEP] rw h, [LAST STEP] apply nat.mul_mod_right, | [STEPS] [STEP] intro h, [LAST STEP] rw nat.mul_mod, |
| Agent Prompt (Reward) | [SUCCESS] [END] | [ERROR MESSAGE] Got error in 'rw h,': error: rewrite tactic failed, did not find instance of the pattern in the target expression x % 2 [END] | [ERROR MESSAGE] Got error in 'apply nat.mul_mod_right,': error: invalid apply tactic, failed to unify x * x % 2 = 0 with ?m_1 * ?m_2 % ?m_1 = 0 [END] | [SUCCESS] [END] |
| ↑ Requests # ——————— ↓ Response # | **Seq # 1** | **Seq # 2** | **Seq # 3** | **Seq # 4** |
| LLM Response | [RUN TACTIC] rw h, [END] | [RUN TACTIC] apply nat.mul_mod_right, [END] | [RUN TACTIC] rw nat.mul_mod, [END] | [RUN TACTIC] rw h, [END] |

Figure 4: We highlight the different parts of the prompts to show how we use the state stack and the execution feedback from the environment. This figure shows the low-level details of the interactions between COPRA and LLM as shown in Figure 1

PROMPTIFY carefully places the different pieces of information relevant to the proof in the prompt. It also includes logic for trimming this information to fit the most relevant parts in the LLM's context window. Every prompt has two parts: the "system prompt" and the "agent prompt."

The agent prompts are synthetically generated using a context-free grammar and contain information about the state stack (including the current proof state), the execution feedback for the previous action, and the set of actions we know to avoid at the current proof state.

The system prompt describes the rules of engagement for the LLM. It contains a grammar (distinct from the one for agent prompts) that we expect the LLMs to follow when it proposes a course of action. The grammar carefully incorporates cases when the response is incomplete because of the LLM's token limits. We parse partial responses to extract the next action using the PARSEACTION routine. PARSEACTION also identifies formatting errors (if any) in the LLM's responses, possibly communicating with the LLM multiple times until these errors are resolved. Figure 4 shows an example back-and-forth between COPRA and LLM, highlighting the low-level details of the use of state stack, execution feedback from ITP, etc.

## 4 EVALUATION

Our findings about COPRA are that: (i) the approach can find proofs significantly quicker than the state-of-the-art finetuning-based baselines, both in terms of number of LLM queries and wall-clock time; (ii) in problems where all current methods fail, COPRA fails faster; (iii) the use of GPT-4, as opposed to GPT-3.5, within the agent is essential for success; and (iv) backtracking significantly improves the system's performance on harder problems. Now we elaborate on our experimental methodology and these results.

**Implementing COPRA.** Our implementation of COPRA can have GPT-3.5, GPT-4, GPT-4-turbo (OpenAI, 2023b) or CodeLlama (Roziere et al., 2023) as the underlying LLM and can interact with both the Lean and the Coq proof environments. Because of the substantial cost of GPT-4 queries,

we cap the number of LLM queries that COPRA can make by 60. To further reduce costs, COPRA first tries to prove its theorems in a single LLM query (one-shot prompting), and then it invokes the agent behavior when it fails to find a proof via one-shot prompting. At first, the retrieval mechanism is not used by the agent to keep the prompts smaller and cost-effective, but if the agent fails to find the proofs then retrieval is used to enrich the proof state before prompting the LLM. More details about the setup can be found in Appendix A.1.1.

The "system prompt" in the one-shot approach is slightly different than that for COPRA, containing instructions to generate a proof in one go rather than step by step. For both COPRA and the one-shot baselines, the prompt contains a single proof example that clarifies how proofs need to be formatted. This proof example remains the same for all test cases.

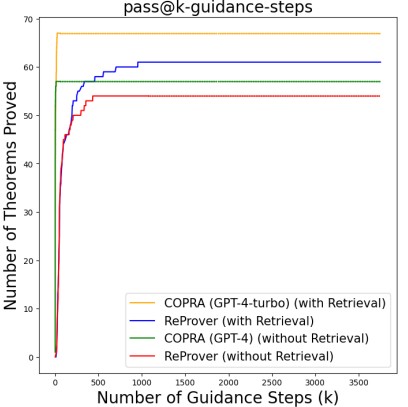

Figure 5: COPRA vs. REPROVER on the miniF2F benchmark

**Benchmarks.** We evaluate our approach on two domains: (i) miniF2F (Zheng et al., 2021), a collection of 244 Lean formalizations of mathematics competition problems, solved using a range of techniques such as induction, algebraic manipulation, and contradiction; and (ii) a set of Coq problems from the CompCert compiler verification project (Leroy, 2009) that was previously used to evaluate the PROVERBOT9001 system Sanchez-Stern et al. (2020).

**Baselines.** We compare with one-shot invocations of GPT-3.5 and GPT-4 in both the miniF2F and the Compcert domains. We also consider an ablation of COPRA that uses GPT-3.5 as its LLM and another that does not use backtracking. Additionally, we also consider GPT-4-turbo, and CodeLLama models for miniF2F domain. For the miniF2F dataset, we also have additional baselines with models like GPT-4-turbo (OpenAI, 2023b) and CodeLlama (Roziere et al., 2023), and ablations with COPRA's retrieval capabilities disabled. As for fine-tuned baselines, a challenge is that all existing open-source theorem-proving systems only target a single-proof environment. As a result, we had to choose different baselines for the Lean (miniF2F) and Coq (Compcert) domains.

Our fine-tuned baseline for the miniF2F domain is REPROVER, a state-of-the-art open-source prover that is part of the Leandojo project (Yang et al., 2023). We use BM25 search on Lean's mathlib library for retrieval of relevant lemmas.

In the Compcert domain, we compare with PROVERBOT9001 (Sanchez-Stern et al., 2020), which, while not LLM-based, is the best publicly available model for Coq. Unlike miniF2F, this benchmark comes with a large training set as well as a test set, and we use the training set for retrieving relevant lemmas and definitions. Our retrieval mechanism, in this case, is a simple BM25 search.

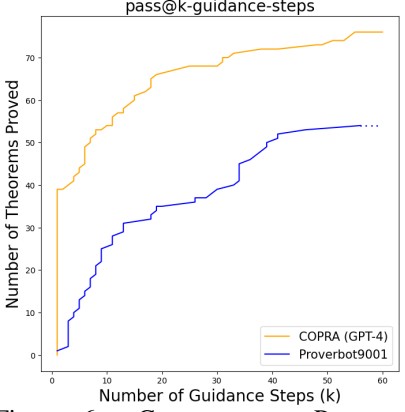

Figure 6: COPRA vs. PROVER-BOT9001 on the Compcert benchmark.

For cost reasons, our evaluation for Compcert uses 118 out the 501 theorems used in the original evaluation of PROVERBOT9001 Sanchez-Stern et al. (2020). For fairness, we include all the 98 theorems proved by PROVERBOT9001 in our subset. The remaining theorems are randomly sampled.

**Metric: pass@$k$-guidance-steps.** The standard metric for evaluating theorem-provers is *pass@k* (Lample et al., 2022; Yang et al., 2023). In this metric, a prover is given a budget of *k proof attempts*; the method is considered successful if one of these attempts leads to success. However, a key objective of our research is to discover proofs *quickly*, with fewer LLM queries and lower wall-clock time. The pass@k metric does not evaluate this characteristic as it does not quantify the number of LLM queries or amount of time needed by a proof attempt.

| Approach | # Theorems proved /# Theorems | % proved | Avg. Guidance Steps in Total | Avg. Guidance Steps on Failure | Avg. Guidance Steps on Pass |
|---|---|---|---|---|---|
| miniF2F Test Dataset | | | | | |
| CodeLlama One Shot | 0/244 | 0.0% | 1 | 1 | 0 |
| GPT 3.5 One Shot | 7/244 | 2.8% | 1 | 1 | 1 |
| COPRA (CodeLlama) | 14/244 | 5.73% | 11.55 | 11.96 | 4.78 |
| GPT 4 One Shot | 26/244 | 10.6% | 1 | 1 | 1 |
| GPT 4-turbo One Shot | 29/244 | 11.88% | 1 | 1 | 1 |
| COPRA (GPT-3.5) (without retrieval) | 29/244 | 11.89% | 12.83 | 14.23 | 2.45 |
| ReProver (without retrieval) | 54/244 | 22.13% | 350.7 | 427.24 | 81.6 |
| COPRA (GPT-4) (without retrieval) | 57/244 | 23.36% | 20.94 | 26.79 | 1.75 |
| ReProver (with retrieval) | 61/244 | 24.9% | 1015.32 | 1312.89 | 122.62 |
| ReProver (with retrieval) (official) | - | 26.5% | - | - | - |
| **COPRA (GPT-4-turbo) (with retrieval)** | **67/244** | **27.45%** | **39.42** | **52.67** | **4.41** |
| CompCert Test Dataset | | | | | |
| GPT 3.5 One-Shot | 10/118 | 8.47% | 1 | 1 | 1 |
| GPT 4 One-Shot | 36/118 | 30.51% | 1 | 1 | 1 |
| Proverbot | **98/118** | **83.05%** | 184.7 | 256.8 | 170.0 |
| **COPRA (GPT-4)** | 76/118 | 64.41% | **12.9** | **10.9** | **16.57** |

Table 1: Aggregate statistics for COPRA and the baselines on `miniF2F` and Compcert

| Approach | Avg. Time In Seconds | | | | | |
|---|---|---|---|---|---|---|
| | Per Proof | | | Per Guidance Step | | |
| | On Pass | On Fail | All | On Pass | On Fail | All |
| ReProver (on CPU) (without retrieval) | 279.19 | 618.97 | 543.78 | 3.42 | 1.45 | 1.55 |
| ReProver (on GPU) (without retrieval) | 267.94 | 601.35 | 520.74 | 2.06 | 0.44 | 0.48 |
| ReProver (on GPU) (with retrieval) | 301.19 | 605.29 | 529.27 | 2.45 | 0.46 | 0.52 |
| COPRA (GPT-3.5) | 39.13 | **134.26** | **122.21** | 15.97 | 9.43 | 9.53 |
| **COPRA (GPT-4) (without retrieval)** | **30.21** | 191.73 | 140.86 | 17.26 | 7.16 | 6.73 |
| COPRA (GPT-4-turbo) (with retrieval) | 68.38 | 598.66 | 450.88 | 15.50 | 11.36 | 11.43 |

Table 2: Average time taken by our approach (COPRA) and ReProver on miniF2F dataset.

To address this concern, we introduce a new metric, *pass@k-guidance*, and evaluate COPRA and its competitors using this metric. Here, we measure the number of correct proofs that a prover can generate with a budget of $k$ *or fewer guidance steps from the LLM or any neural model*. For LLMs, one guidance step is a single inference query. One challenge here is that we want this metric to be correlated with the number of correct proofs that the prover produces within a wall-clock time budget; however, the cost of an inference query is proportional to the number of responses generated per query. To maintain the correlation between the number of inference queries and wall-clock time, we restrict each inference on LLM to a single response. (more details about the metric is in Appendix A.1.3)

**Results** Figure 5 shows our comparison results for the `miniF2F` domain. As we see, COPRA outperforms REPROVER, completing, within just 60 guidance steps, problems that REPROVER could not solve even after a thousand guidance steps. This is remarkable given that COPRA is based on a black-box foundation model and REPROVER was fine-tuned for at least a week on a dataset derived from Lean's Mathlib library. For fairness, we ran REPROVER multiple times with 16, 32, and 64 (default) as the maximum number of guidance steps per proof step. We obtained the highest success rates with 64 guidance steps.

Figure 6 shows a comparison between COPRA and PROVERBOT9001.

| Approach | # Theorems proved /# Theorems | % proved |
|---|---|---|
| miniF2F Test Dataset | | |
| COPRA (GPT-4) w/o backtracking | 56/244 | 22.95% |
| **COPRA (GPT-4)** | **57/244** | **23.36%** |
| CompCert Test Dataset | | |
| COPRA (GPT-4) w/o backtracking | 52/118 | 44.06% |
| **COPRA (GPT-4)** | **76/118** | **64.41%** |

Table 3: Ablation showing the effectiveness of backtracking

We find that COPRA is significantly faster than PROVERBOT9001. Since we put a cap of 60 guidance steps on COPRA, it cannot prove all the theorems that PROVERBOT9001 eventually proves. However, as shown in the figure, COPRA proves many more theorems than PROVERBOT9001 if only 60 guidance steps are allowed. Specifically, we prove 77.5% of the proofs found by PROVERBOT9001 in less than 60 steps.

Aggregate statistics for the two approaches, as well as a comparison with the one-shot GPT-3.5 and GPT-4 baselines (details of baseline setup are mentioned in Appendix A.1.2), appear in Table 1. It is clear from this data that the language-agent approach offers a significant advantage over the one-shot approach. For example, COPRA solves more than twice as many problems as the one-shot GPT-4 baseline, which indicates that it does not just rely on GPT-4 recalling the proof from its memory (we discuss this in more details in Appendix A.1.5). Also, the use of GPT-4 as opposed to GPT-3.5 seems essential.

```
theorem algebra_sqineq_at2malt1
(a : ℝ) :
a * (2 - a) ≤ 1 :=
begin
    have h : ∀ (x : ℝ), 0 ≤ (1 - x) ^ 2,
    from λ x, pow_two_nonneg (1 - x),
    calc a * (2 - a)
            = 1 - (1 - a) ^ 2 : by ring
        ... ≤ 1 : sub_le_self _ (h a),
end
```

Figure 7: A theorem in the 'algebra' category that CO-PRA could prove but REPROVER could not.

We establish the correlation between the number of guidance steps needed for a proof and wall-clock time in Table 2 (more details are discussed in Appendix A.1.4). Although the average time per guidance step is higher for COPRA, COPRA still finds proofs almost 9x faster than REPROVER. This can explained by the fact that our search is more effective as it uses 46x fewer guidance steps than REPROVER. These guidance steps not only contain the average time spent on generating responses from LLM but at times have some contribution corresponding to the execution of the tactic on the Lean environment itself.

Table 2 also offers data on when the different approaches report failures. Since REPROVER uses a timeout for all theorems, we also use a timeout of 10 minutes while considering failures in Table 2. The data indicates that COPRA is comparatively better at giving up when the problem is too hard to solve. We also note that less time is spent per guidance step in case of failure for all approaches.

We show the impact of ablating the backtracking feature of COPRA in Table 3. We note that backtracking has a greater positive impact in the Compcert domain. We hypothesize that this is because the Compcert problems are more complex and backtracking helps more when the proofs are longer.

Finally, we offer an analysis of the different categories of `miniF2F` problems solved by COPRA and REPROVER in Figure 8. We see that certain kinds of problems, for example, International Mathematics Olympiad (IMO) problems and theorems that require induction, are difficult for all approaches. However, Figure 8b shows that COPRA takes fewer steps consistently across various categories of problems in `miniF2F`.

From our qualitative analysis, there are certain kinds of problems where the language-agent approach seems especially helpful. For instance, Figure 7 shows a problem in the 'algebra' category that REPROVER could not solve. More examples of interesting Coq and Lean proofs that COPRA found appear in the appendix.

## 5 RELATED WORK

**Supervised Learning for Theorem-Proving.** There is a sizeable literature on search-based theorem-proving techniques based on supervised learning. These methods train a model to predict the next proof step at each point in a proof. This model is then used to guide a search technique, e.g., best-first or depth-limited search, that synthesizes a proof. Earlier methods of this sort used small-scale neural networks (Yang & Deng, 2019; Sanchez-Stern et al., 2020; Huang et al., 2019) as predictors. More recent methods, such as GPT-f (Polu & Sutskever, 2020), PACT (Han et al., 2021), HyperTree Proof Search (Lample et al., 2022), and REPROVER (Yang et al., 2023), have used LLMs. COPRA has some resemblance with the latter approaches. However, it departs from these prior methods in using execution feedback and a more sophisticated search algorithm.

The recent Draft-Sketch-Proof (Jiang et al., 2022) method relies on informal proofs to generate formal proofs. Other methods like Baldur (First et al., 2023) generate the whole proof in one shot using an LLM and then *repair* it. The main ideas in these efforts — the use of informal proofs and repair models — are orthogonal to our approach (we discuss this in more detail in Appendix A.1.6).

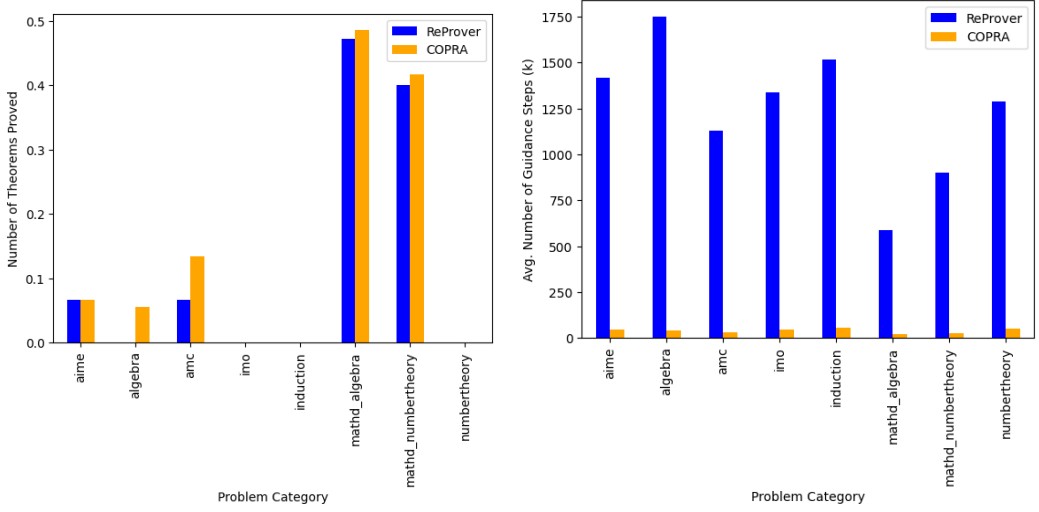

(a) Problems solved in different categories

(b) Number of guidance steps in different categories

Figure 8: Breakdown of theorems proved in various categories

**Reinforcement Learning for Theorem-Proving.** Kaliszyk et al. (2018) pioneered the use of RL in theorem-proving; subsequently, Wu et al. (2021) gave TacticZero, a deep RL approach to the problem. TacticZero does not use LLMs, thus missing out on a key source of generic mathematical knowledge. Also, COPRA has retrieval capabilities that TacticZero lacks.

**Language Agents.** Several distinct LLM agent architectures have been proposed over the last year (Significant-Gravitas, 2023; Yao et al., 2022; Shinn et al., 2023; Wang et al., 2023). These models combine an LLM's capability to use tools Schick et al. (2023), decompose a task into subtasks (Wei et al., 2022; Yao et al., 2023), and self-reflect (Shinn et al., 2023) However, we are the first to offer an LLM agent for theorem-proving. We also distinguish ourselves from prior work along these lines by introducing a more efficient stateful search in the policy.

# 6 CONCLUSION

We have presented COPRA, the first LLM-agent approach to formal theorem-proving. The approach departs from prior LLM-based theorem-proving techniques by explicitly modeling the interaction between the prover and the proof environment. It also goes beyond prior language-agent approaches for any domain in using a stateful backtracking search within the policy.

Many questions remain open. First, we gave our GPT-4 a budget of a maximum of 60 inferences per problem for cost reasons. Whether the learning dynamics would drastically change with a much larger inference budget remains to be seen. A related question is whether a GPT-4-scale model is truly essential for our task. We have shown that the cheaper GPT-3.5 agent is not competitive against our GPT-4 agent; however, it is possible that a different, more affordable foundation model would have done better. Finally, our proof MDP also enables approaches where an LLM policy is fine-tuned using RL. It remains to be seen how such an approach, done by necessity with smaller-scale models, would compare with our in-context-learning approach.

# 7 REPRODUCIBILITY STATEMENT

We are releasing all the code needed to run COPRA as supplementary material. The code contains all "system prompts" described in Section A.4 and Section A.3, along with any other relevant data needed to run COPRA. However, to use our code, one must use their own OpenAI API keys. An issue with reproducibility in our setting is that the specific models served via the GPT-4 and GPT-

3.5 APIs may change over time. In our experiments, we set the "temperature" parameter to zero to ensure the LLM outputs are as deterministic as possible.

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

# A    APPENDIX

## CONTENTS

| Approach | # Theorems proved /# Theorems | % proved | Avg. Guidance Steps in Total | Avg. Guidance Steps on Failure | Avg. Guidance Steps on Pass |
|---|---|---|---|---|---|
| miniF2F Test Dataset | | | | | |
| GPT 4-turbo One Shot | 29/244 | 11.88% | 1 | 1 | 1 |
| COPRA (GPT-4-turbo) (agent + retrieval) | 56/244 | 22.95% | 18.75 | 23.63 | 2.35 |
| COPRA (GPT-4-turbo) (agent) | 60/244 | 24.59% | 21.64 | 27.47 | 3.76 |
| COPRA (GPT-4-turbo) (agent + one-shot) | 62/244 | 25.40% | 22.58 | 28.57 | 3.70 |
| **COPRA (GPT-4-turbo) (agent + one-shot + retrieval)** | **67/244** | **27.45%** | **39.42** | **52.67** | **4.41** |

Table 4: Aggregate statistics for COPRA capabilities and COPRA ensemble on `miniF2F`

## A.1 EVALUATION DETAILS

### A.1.1 COPRA IMPLEMENTATION SETUP DETAILS

We introduce a common proof environment for COPRA, which can also be used by any other approach for theorem proving. The proof environment is agnostic of language and domain, having a common interface that makes COPRA work seamlessly for both Lean and Coq. As per our knowledge, this is the first language and domain-agnostic interface that can allow training or testing of various neural theorem-proving approaches. In the future, we plan to support more proof languages. We also have support for various LLMs other than GPTs, including open LLMs like Llama 2 (Touvron et al., 2023), Code Llama (Roziere et al., 2023), etc. All the theorems are searched within a timeout of 10 minutes and with a maximum of 60 LLM inference calls (whichever exhausts first). To make it comparable across various LLMs, only one response is generated for one inference. All these responses are generated with the *temperature* set to zero, which ensures that the responses generated are more deterministic, focussed, and comparable.

We use GPT-3.5, GPT-4, GPT-4-turbo (OpenAI, 2023b), and CodeLLama (Roziere et al., 2023) to test the capabilities of COPRA. We find that it is best to use COPRA's different capabilities in an ensemble, which makes it not only more accurate but enhances its performance. Therefore, we first use one-shot prompting to find the proof, then we use COPRA without retrieval upon failure and then run COPRA with retrieval only when we fail again. To ensure fairness in comparison, we make sure that the number of guidance steps is capped at 60 and the 10-minute timeout is spread across all these three executions. From Table 4, it is clear that despite the significant overlap between the three executions, the ensemble covers more cases. One possible reason could be that the addition of extra information from retrieval can sometimes be misleading because the retriever is not perfect and it can find lemmas that are not completely relevant to proving the goal. Nevertheless, sometimes these extra lemmas are handy, so we can best use the different capabilities as an ensemble.

### A.1.2 ONE-SHOT BASELINE SETUP DETAILS

We run the one-shot GPT-4 baseline by calling the LLM exactly once. Additional queries are only used when the response is incomplete or ill-formatted. To ensure a fair comparison of one-shot baseline with GPT-4 COPRA agent with 60 inference calls allowed, we always set the *temperature* parameter as zero for all LLM queries.

### A.1.3 METRIC: *pass@k-guidance-steps*

The main motivation behind the *pass@k-guidance-steps* is to assess the speed of the proposed approach and the effectiveness of the LLM or neural network to guide the proof search. It is a reasonable metric because it does a more even-handed trade-off in accounting for the time taken to complete a proof and at the same time ignores very low-level hardware details.

Different approaches need a different amount of guidance from a neural model to find the right proof. For example, approaches like Baldur (First et al., 2023), DSP (Jiang et al., 2022), etc., generate the whole proof all at once. On the other hand, GPT-f (Polu & Sutskever, 2020), PACT

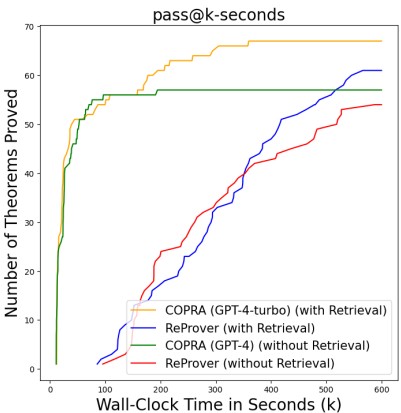

Figure 9: COPRA vs. REPROVER on the `miniF2F` benchmark

(Han et al., 2021), REPROVER (Yang et al., 2023), Proverbot (Sanchez-Stern et al., 2020), or our approach generate the proofs step by step. We argue that *pass@k-guidance-steps* is a fairer metric to compare these different types of approaches because it correlates with the effectiveness of the proof-finding algorithm in an implementation-agnostic way. Since the exact time might not always be a good reflection of the effectiveness because of hardware differences, network throttling, etc., it makes sense to not compare directly on metrics like pass@k-minutes or pass@k-seconds. Not only these metrics will be brittle and very sensitive to the size, hardware, and other implementation details of the model, but not every search implementation will be based on a timeout. For example, Proverbot does not use timeout-based search (and hence we don't compare on the basis of time with Proverbot9001).

### A.1.4 *pass@k-guidance-steps* VERSUS WALL-CLOCK TIME

We show that *pass@k-guidance*, correlates very well with wall-clock time for finding proofs by using the metric *pass@k-seconds*. *pass@k-seconds* measures the number of proofs that an approach can find in less than $k$ seconds. The plot in Figure 9 shows that *pass@k-seconds* follows the same trend as *pass@k-guidance-steps* as shown in Figure 5.

We can use the comparison of COPRA with REPROVER (Yang et al., 2023) on the miniF2F dataset to explain the correlation between finding proofs fast and *pass@k-guidance-steps*. From Table 2, we know that on average the time taken per guidance (which includes time taken to execute the proof steps on ITP as well) is around 1.55 seconds for REPROVER and 6.73 seconds for COPRA. Given that REPROVER's guidance LLM is small, we can assume that REPROVER doesn't take any time (zero time) to query its LLM and spends most of the 1.55 seconds running the proof steps on ITP. Now, we can reasonably estimate GPT-4 average response time to be 5 seconds (6.73 - 1.55) from Table 2. However, we see that the number of guidance used by REPROVER is about 46x higher on success. This interestingly shows up in the wall clock time too which is around 9x higher ( 46x/5) for REPROVER on success, so there is a tradeoff between the two, but the number of guidance steps dominates when the guidance model is not good. So, if the guidance model is good (it may be as big as GPT), we can empirically argue that asymptotically the search will converge to proof faster (given that it can be found using that guidance model).

### A.1.5 DATA LEAKAGE IN GPT-4

With LLM pretraining data getting larger and larger, it is hard to know if there is any accidental leakage of the evaluation set in the training data of the LLM itself. The data leakage problem is applicable to all generative AI approaches based on large pretrained models, whose pretraining data is rarely publicly accessible. For coding and language generation tasks, which have been studied in more depth, the use of large pretrained LLMs has now become standard, simply because the benefits of scale are simply too significant to ignore. We believe that AI-for-math is also taking a similar trajectory.

Data leakages can be direct or indirect hints to solve the evaluation set. Even with open LLMs like Llama (Touvron et al., 2023), it is computationally hard to detect these hints in the pertaining data given the LLMs are trained on trillions of tokens. However, after a thorough analysis of the proofs generated by COPRA on the miniF2F dataset, we can safely conclude that data leakage isn't a significant contributor to our results, for several reasons.

First, we note that COPRA significantly outperforms one-shot invocations of GPT-4. If the results on COPRA were significantly tainted by data leakage, we would have expected better performance from one-shot GPT-4.

Second, not all the formal proofs of the miniF2F test dataset are available online (only 80 proofs are available in Lean). It is highly unlikely that GPT-4 has been trained on proof-state and tactic pair generated by hooking up the Lean Interactive Theorem Prover (ITP). Moreover, since the ground truth of `miniF2F` test for Lean is still not available, even if it were trained on proof-states one still needs to manually annotate ground truth tactics. Given that GPT-4 is a general-purpose LLM, it is highly unlikely that while training GPT-4 the miniF2F dataset was first manually annotated, and then proof-state and tactic pair information was collected by hooking up the Lean ITP.

Also, in our agent interactions, we limit ourselves only to the goal at that point. There is no mention of the original theorem anywhere (except for the very first proof-state), so the chances that GPT-4 can correlate any intermediate state with the original theorem are very low unless it can somehow manage to simulate Lean's interactive theorem proving within itself. It is also improbable that GPT-4 has seen the proof-state in the same format that we use, let alone using the execution feedback which has not been used in any known previous works.

One could hypothesize that some of the one-shot GPT-4 proofs might be influenced by potential training on the miniF2F dataset. However, this doesn't seem to be true because we see that most of the proofs we generated were either not mentioned in the miniF2F test dataset or completely different from the manually written proofs in the miniF2F test dataset (including the first step mismatch). Table 5 shows the detailed analysis of proofs generated by COPRA and the proofs mentioned in $miniF2F$ test dataset for Lean. From the Table 5, it is clear that most of the proofs generated by COPRA are different from the proofs mentioned in the $miniF2F$. The ones that are exactly the same are simple single-tactic proofs that just use exactly one of the `linarith`, `nlinarith`, or `norm_num` tactics without any arguments. If we set aside these straightforward simple cases, then about $92\%$ of the proofs generated by COPRA are either different from the proofs mentioned in the `miniF2F` or do not have a proof mentioned in the `miniF2F` dataset. Out of all proofs generated by COPRA about 25.37% proofs are for theorems that have no proofs mentioned in the `miniF2F` test dataset as compared to 22.95% for REPROVER. Some of the proofs generated by our approach as compared to proofs mentioned in the `miniF2F` test dataset are shown in Figure 10.

Finally, the ability of agent interactions to enhance the basic LLM approach seems to transcend OpenAI's LLMs. We ran COPRA on the recently released CodeLLama LLM. From Table 1, COPRA improved CodeLlama's capabilities to prove theorems by about $5\%$ on `miniF2F` dataset. This indicates that the in-context learning capabilities that we build are transferable and LLM-agnostic.

### A.1.6 COMPARISON WITH METHODS USING INFORMAL PROOFS

A formal proof is something that can be automatically verified using an Interactive Theorem Prover (ITP), whereas an informal proof can only be verified by a human. ITP is a software tool to assist with the development of formal proofs by human-machine collaboration. This involves some sort of interactive proof editor, or other interfaces, with which a human can guide the search for proofs. Often formal proofs are much more rigorous and pedantic than informal proofs. So informal proof can be loosely considered as a proof sketch based on which one can write rigorous machine-checkable formal proofs.

Methods that use DSP (Jiang et al., 2022) pipeline that uses informal proofs to find the formal proofs work very well on datasets like `miniF2F` which have problems from math competitions. However, real-world math formulations are not necessarily math competition problems with well-known informal proofs. Certain domains like software verification like CompCert don't have any notion of informal proofs. It is important to note that having access to informal proofs (human-written or LLM-generated) simplifies the problem of synthesizing the formal proof into more of a translation problem, and that is one of the reasons why DSP-like approaches perform well on

| | Proofs in $miniF2F$ | | | | | | Proofs **NOT** in $miniF2F$ | Total |
|---|---|---|---|---|---|---|---|---|
| | Single-Tactic Simple Proofs | | | Two-Tactic Proofs | Longer OR Complex Proofs | Total | | |
| Tactics Used ——— Proof Count | **linarith** | **norm_num** | **nlinarith** | two tactics | > 2 tactics OR 1 tactic multi-args | | **sorry** | |
| $miniF2F$ Proof Count | 11 | 12 | 2 | 16 | 39 | 80 | 164 | 244 |
| Exact Match COPRA Count | 7 | 9 | 1 | 4 | 0 | 21 | 0 | 21 |
| $1^{st}$ Tactic Match COPRA Count | 7 | 9 | 1 | 7 | 1 | 25 | 0 | 25 |
| Distinct COPRA Count | 2 | 3 | 1 | 8 | 15 | 29 | 17 | 46 / 67 **68.65%** |
| Distinct COPRA Count ex Single-Tactic | - | - | - | 8 | 15 | 29 | 17 | 46 / 50 **92%** |
| All COPRA Count | 9 | 12 | 2 | 12 | 15 | 50 | 17 | 67 |

Table 5: Analysis of proof generated by COPRA on `miniF2F` test dataset for Lean.

`miniF2F` dataset. These approaches will work very well if the LLM can use its memory component to effectively recall the informal proof of a well-known math competition problem. However, a lot of formal theorem proving happens on a completely new set of formalizations which are heavily customized depending on the domain. For example, in the field of software verification, we will have a custom mathematical model defined for the software which has no informal proofs or definitions which can be recalled from memory by the LLMs. It is also important to note that the accuracy numbers in (Jiang et al., 2022) are not directly comparable to ours because they use different proof languages. Isabelle is used in DSP-like approaches which have powerful automatic reasoning tools like Sledgehammer, unlike Lean.

Our approach is generic and can work for any domain where formal theorem proving is used. It is one of the first methods which is domain and proof language agnostic.

## A.2 EXAMPLE PROOFS GENERATED FOR MINIF2F

Fig. 11 shows some other interesting proofs generated by our approach on miniF2F dataset.

## A.3 SYSTEM PROMPTS FOR MINIF2F

Parts of the 'system prompt' used by COPRA for theorem proving in Lean are shown in Fig. 12.

```
(a.1)

theorem algebra_sqineq_at2malt1
  (a : ℝ) :
  a * (2 - a) ≤ 1 :=
begin
  suffices: 0 ≤ a^2 - 2*a + 1, nlinarith,
  suffices: 0 ≤ (a - 1)^2, nlinarith,
  nlinarith,
end

(b.1)

theorem mathd_algebra_270
  (f : ℝ → ℝ)
  (h₀ : ∀ x ≠ -2, f x = 1 / (x + 2)) :
  f (f 1) = 3/7 :=
begin
  rw [h₀, h₀],
  norm_num,
  linarith,
  rw h₀,
  norm_num,
  linarith,
end

(c.1)

theorem mathd_numbertheory_229 :
  (5^30) % 7 = 1 :=
begin
  have five_to_thirty_is_one :
    (5^30 : zmod 7) = 1 :=
  begin
    have five_to_the_six_is_one : (5^6 :
    zmod 7) = 1, by dec_trivial,
    have break_power : (5^30 : zmod 7) =
    (5^6)^5, by norm_num,
    rw break_power,
    rw five_to_the_six_is_one,
    norm_num,
  end,
  change 5^30 ≡ 1 [MOD 7],
  rw ←zmod.eq_iff_modeq_nat,
  exact_mod_cast five_to_thirty_is_one,
end
```

```
(a.2)

theorem algebra_sqineq_at2malt1
(a : ℝ) :
a * (2 - a) ≤ 1 :=
begin

    -- We will complete the square to
    show that the expression is at most
    1.
  have h : a * (2 - a) = 1 - (1 - a)^2,
  { ring },
  rw h,
  -- Now we show that (1 - a)^2 is non-
    negative, which implies 1 - (1 - a)^2
    is at most 1.
  apply sub_le_self,
  apply pow_two_nonneg,
end

(b.2)

theorem mathd_algebra_270
(f : ℝ → ℝ)
(h₀ : ∀ x ≠ -2, f x = 1 / (x + 2)) :
f (f 1) = 3/7 :=
begin
    have h₁ : f 1 = 1 / (1 + 2) := h₀ 1 (
    by linarith),
    rw h₁,
    rw h₀,
    field_simp,
    ring,
    apply ne_of_gt,
    norm_num,
end

(c.2)

theorem mathd_numbertheory_229 :
(5^30) % 7 = 1 :=
begin
    norm_num,
end
```

Figure 10: Some proofs found by COPRA as compared to the proofs mentioned in the miniF2F test dataset. It is interesting to see that the proofs generated by COPRA are different from the proofs mentioned in the repository. This is especially true when the proofs are longer. It is also worth noting that occasionally COPRA can find very simple proofs for longer proofs mentioned in miniF2F test dataset. (a.1), (b.1), and (c.1) show the proofs as mentioned in the miniF2F dataset, while (a.2), (b.2), and (c.2) show the corresponding proofs generated by COPRA.

```
(a)
theorem
    mathd_algebra_246
(a b : ℝ)
(f : ℝ → ℝ)
(h₀ : ∀ x, f x = a * x^4
    - b * x^2 + x + 5)
(h₂ : f (-3) = 2) :
f 3 = 8 :=
begin
    rw h₀,
    rw h₀ at h₂,
    ring_nf,
    linarith,
end

(b)
theorem
mathd_numbertheory_247
(n : ℕ)
(h₀ : (3 * n) % 2 = 11)
    :
n % 11 = 8 :=
begin
    cases (nat.
mod_two_eq_zero_or_one
(3 * n)),
    exfalso,
    linarith,
    linarith,
end
```

```
(c)
theorem mathd_algebra_44
(s t : ℝ)
(h₀ : s = 9 - 2 * t)
(h₁ : t = 3 * s + 1) :
s = 1 ∧ t = 4 :=
begin
    split,
    {
        rw h₀ at h₁,
        linarith
    },
    rw h₀ at h₁,
    rw h₁ at h₀,
    linarith,
end

(d)
theorem amc12b_2002_p2
(x : ℤ)
(h₀ : x = 4) :
(3 * x - 2) * (4 * x +
    1) - (3 * x - 2) *
    (4 * x) + 1 = 11 :=
begin
    ring_nf,
    rw h₀,
    ring,
end
```

```
(e)
theorem mathd_algebra_107
(x y : ℝ)
(h₀ : x^2 + 8 * x + y^2 - 6 *
    y = 0) :
(x + 4)^2 + (y-3)^2 = 5^2 :=
begin
    rw pow_two at h₀,
    rw add_assoc at h₀,
    rw add_comm (x * x) (8 *
    x + y ^ 2) at h₀,
    rw add_comm (8 * x) (y ^
    2) at h₀,
    rw add_assoc at h₀,
    rw add_comm (y ^ 2) (8 *
    x + x * x) at h₀,
    rw add_assoc at h₀,
    ring_nf at h₀,
    rw pow_two,
    rw pow_two,
    ring_nf,
    rw ←add_assoc,
    rw h₀,
    linarith,
end
```

Figure 11: Some other interesting proofs generated for miniF2F by COPRA. The length of the proofs generated shows that interaction with the environment helps in fixing the errors encountered while writing long proofs. These long sequences of rewrites are not easy to synthesize without knowing the exact execution feedback from the environment which often contains the hint to fix the rewrites.

You are a proficient formal theorem-proving agent in Lean 3. You can predict
↪   the next proof step given the current proof state. The proof state is
↪   described in the following format:
**1.** All the goals are described under `[GOALS]` keyword. Each goal within
↪   the `[GOALS]` is described under the keyword `[GOAL] i`, where `i` is a
↪   positive integer. For example, `[GOAL] 1`, `[GOAL] 2`, etc.
**2.** Within each `[GOAL] i` keyword, the goal is described as a human-readable
↪   serialized version of the proof state as shown while running `lean`
↪   command. Each goal, might also accompany some hypotheses, which are
↪   described under the keyword `[HYPOTHESES] i`. Each hypothesis within
↪   `[HYPOTHESES]`, starts with the prefix `[HYPOTHESIS]`.
**3.** Sometimes `[GOALS]` can have description about the proof state like
↪   `Proof finished`, `There are unfocused goals`, `Not in proof mode`,
↪   etc. The description is described under the keyword `[DESCRIPTION]`.
**4.** Finally, `[STEPS]` keyword is used to describe proof-steps used so far.
↪   Each proof step starts with the prefix `[STEP]`, and is a valid Lean
↪   tactic. For example, `[STEPS][STEP]rw $h_1$ at $h_2$,[STEP]{linarith},`.
**5.** Sometimes, `[INCORRECT STEPS]` keyword optionally used to describe
↪   proof-steps which should NOT be generated. Use this as a hint for not
↪   generating these proof-steps again as they failed previously. For
↪   example, `[INCORRECT STEPS][STEP]apply $h_1$,[STEP]rw ←$h_1$`.
**6.** There is also an optional `[LAST STEP]` keyword which describes the
↪   proof-step generated last time. If the proof-step was incorrect, then
↪   it is also followed by error message from Coq environment. For example,
↪   `[LAST STEP]linarith,\n[ERROR MESSAGE]linarith failed to find a
↪   contradiction\nstate:\nx y : $\mathbb{R}$,\nh$h_1$ : x = 3 - 2 * y,\nh$h_2$ : 2 * x - y =
↪   1\nh⊢ false`. If the proof-step was correct then it is followed by the
↪   keyword `[SUCCESS]`. For example, `[LAST STEP]linarith,[SUCCESS]`.
↪   Don't generate the last proof-step again if it was NOT successful.
**7.** Sometimes there can be errors in the format of the generated response.
↪   This is reported using the keyword `[ERROR]` followed by the error
↪   message. For example, `[ERROR]\nInvalid response:\n'Great! The proof is
↪   complete.', \nStopping Reason: 'stop'.\n Please respond only in the
↪   format specified.[END]`. This means that the response generated by you
↪   was not in the specified format. Please follow the specified format
↪   strictly.

If you think you know the next proof step, then start your response with
↪   `[RUN TACTIC]` followed by the next proof-step which will help in
↪   simplifying the current proof state. For example, `[RUN
↪   TACTIC]induction c,[END]`. Generate exactly ONE proof-step. Multiple
↪   proof steps are more error prone, because you will not get a chance to
↪   see intermediate proof state descriptions. Make sure that the proof
↪   step is valid and compiles correctly in Lean 3.

You can refer to the example conversation to understand the response format
↪   better. It might also contain some similar proof states and their
↪   corresponding proof-steps.

Please take a note of the following:
**1.** Make sure to end all your responses with the keyword `[END]`. Follow the
↪    specified format strictly.
**2.** While generating `[RUN TACTIC]` keyword, do NOT generate the tactics
↪    mentioned under `[INCORRECT STEPS]`......
..............

Figure 12: Parts of 'system prompt' used by COPRA for Lean

## A.4 SYSTEM PROMPTS FOR COMPCERT

Parts of the 'system prompt' used by COPRA for theorem proving in Coq are shown in Fig. 13.

## A.5 EXAMPLE PROOFS GENERATED FOR COMPCERT

Fig. 14 shows some interesting proofs generated by our approach on the CompCert dataset.

```
You are a proficient formal theorem-proving agent in Coq. You can predict
↪  the next proof step given the current proof state, relevant
↪  definitions, and some possible useful lemmas/theorems. The proof state
↪  is described in the following format:
1. All the goals are described under `[GOALS]` keyword. Each goal within
↪  the `[GOALS]` is described under the keyword `[GOAL] i`, where `i` is a
↪  positive integer. For example, `[GOAL] 1`, `[GOAL] 2`, etc.
2. Within each `[GOAL] i` keyword, the goal is described as a human-readable
↪  serialized version of the proof state as shown while running `coqtop`
↪  command. Each goal, might also accompany some hypotheses, which are
↪  described under the keyword `[HYPOTHESES] i`. Each hypothesis within
↪  `[HYPOTHESES]`, starts with the prefix `[HYPOTHESIS]`. Apart from the
↪  goal and hypothesis, some OPTIONAL keywords like `[DEFINITIONS] i` and
↪  `[THEOREMS] i` are also present which describe the relevant definitions
↪  of symbols used in that goal, and some possible useful theorems or
↪  lemmas which might help in simplifying the goal. Each definition within
↪  `[DEFINITIONS]` starts with the prefix `[DEFINITION]`. Similarly, each
↪  theorem/lemma under `[THEOREMS]` keyword starts with the prefix
↪  `[THEOREM]`. These definitions and theorems can be used to simplify the
↪  goal using the tactics like rewrite, apply, etc. However, it is also
↪  possible that these definitions and theorems are not used at all.
3. Sometimes `[GOALS]` can have description about the proof state like
↪  `Proof finished`, `There are unfocused goals`, `Not in proof mode`,
↪  etc. The description is described under the keyword `[DESCRIPTION]`.
4. Finally, `[STEPS]` keyword is used to describe proof-steps used so far.
↪  Each proof step starts with the prefix `[STEP]`, and is a valid Coq
↪  tactic ending with a `.`. For example, `[STEPS][STEP]intros
↪  a.[STEP]induction a.`.
5. Sometimes, `[INCORRECT STEPS]` keyword optionally used to describe
↪  proof-steps which should NOT be generated. Use this as a hint for not
↪  generating these proof-steps again as they failed previously. For
↪  example, `[INCORRECT STEPS][STEP]apply mul_assoc.[STEP]rewrite <- H.`.
6. There is also an optional `[LAST STEP]` keyword which describes the
↪  proof-step generated last time. If the proof-step was incorrect, then
↪  it is also followed by error message from Coq environment. For example,
↪  `[LAST STEP]reflexivity.[ERROR MESSAGE]Error: In environment\nn :
↪  nat\nUnable to unify "n" with "n + 0".`. If the proof-step was correct
↪  then it is followed by the keyword `[SUCCESS]`. For example, `[LAST
↪  STEP]reflexivity.[SUCCESS]`. Don't generate the last proof-step again
↪  if it was NOT successful.
7. Sometimes there can be errors in the format of the generated response.
↪  This is reported using the keyword `[ERROR]` followed by the error
↪  message. For example, `[ERROR]\nInvalid response:\n'Great! The proof is
↪  complete.', \nStopping Reason: 'stop'.\n Please respond only in the
↪  format specified.[END]`. This means that the response generated by you
↪  was not in the specified format. Please follow the specified format
↪  strictly.

If you think you know the next proof step, then start your response with
↪  `[RUN TACTIC]` followed by the next proof-step which will help in
↪  simplifying the current proof state. For example, `[RUN TACTIC]destruct
↪  c.[END]`. Generate exactly ONE proof-step. Multiple proof steps are
↪  more error prone, because you will not get a chance to see intermediate
↪  proof state descriptions. Make sure that the proof step is valid and
↪  compiles correctly with Coq.
......................
```

Figure 13: Parts of 'system prompt' used by COPRA for Coq

(a)

```
gss :
forall l v m,
(set l v m) l = match l with R
↪  r =>
    v | S sl ofs ty =>
    Val.load_result
    ↪  (chunk_of_type ty) v
    ↪  end.
Proof.
    intros l v m.
    destruct l as [r | s o t].
    - unfold set.
    destruct (Loc.eq (R r) (R
    ↪  r)); [reflexivity |
    ↪  contradiction].
    - unfold set.
    destruct (Loc.eq (S s o t)
    ↪  (S s o t));
    ↪  [reflexivity |
    ↪  contradiction].
Qed.
```

(b)

```
eq : forall (p q: loc), {p =
↪  q} + {p <> q}.
Proof.
    decide equality.
    - apply mreg_eq.
    - decide equality.
    - decide equality.
    apply Pos.eq_dec.
    decide equality.
    - decide equality.
Qed.
```

(c)

```
disjoint_cons_right
↪  :
forall a l1 l2,
disjoint l1 (a ::
↪  l2) -> disjoint
↪  l1 l2.
Proof.
    intros a l1 l2
    ↪  H.
    unfold
    ↪  disjoint.
    intros x1 x2 H1
    ↪  H2.
    apply H.
    assumption.
    right.
    assumption.
Qed.
```

(d)

```
eq_int_type :
↪  forall (x y:
↪  int_type),
↪  {x=y} + {x<>y}.
Proof.
    decide
    ↪  equality.
Qed.
```

(e)

```
set_locals_lessdef
↪  :  forall e1
↪  e2,
↪  env_lessdef e1
↪  e2 -> forall
↪  il,
↪  env_lessdef
↪  (set_locals il
↪  e1)
↪  (set_locals il
↪  e2).
Proof.
    intros e1 e2 H.
    induction il as
    ↪  [| a il'].
    - apply H.
    - intros.
    apply
    ↪  set_var_lessdef.
    apply IHil'.
    apply
    ↪  Val.lessdef_refl.
Qed.
```

Figure 14: Some other interesting proofs generated for CompCert by COPRA. We can see that these proofs are long, and often use 'apply' tactic which shows that COPRA can effectively use the retrieved information to discharge the current proof state.

