# OpenReview forum: "A Language-Agent Approach to Formal Theorem-Proving"
_ICLR.cc/2024/Conference — Submitted to ICLR 2024_

### Official Review · Reviewer_4t7K · 2023-10-28

**Soundness:** 3 good
**Presentation:** 2 fair
**Contribution:** 2 fair
**Rating:** 5
**Confidence:** 4

**Summary:**

This paper introduces COPRA, an approach to theorem proving that uses off-the-shelf, high-capacity LLM (GPT-4 in this case) as part of a policy that interacts with a proof environment.
At each step, the policy consumes a textual prompt by using the underlying proof assistant, or backtrack, or retrieve relevant lemmas and definitions from an external corpus.
The feedback from the execution is used toconstruct a new prompt for the policy, and the process reiterates.
The proposed approach is evaluated empirically.

**Strengths:**

1) The problem is rigorously formalised as a Markov decision process in reinforcement learning.

2) The proposed approach compares favourably wrt the state of the art, but the differences between the different baselines make the comparison a bit opaque.

**Weaknesses:**

1) The problem is formalised as an RL task, but then the authors say "In this paper, we do not take on this problem. Instead, we consider a fixed policy - a wrapper around a pretrained LLM (GPT-4) that can learn in-context - and show that this policy can achieve a high reward".
See question below.

2) The structure of the framework proposed is not very clear. The algorithm in Fig. 3 is quite high-level. The calls and interactions with the LLMs are not discussed in much detail.

3) on p. 9, the authors say: "However, [COPRA] departs from these prior methods in using execution feedback and a more sophisticated search algorithm."
It is not clear to me what this more sophisticate search algorithm is, specifically why it is sophisticate.

**Questions:**

1) Why defining the problem as an RL task, if this is not used in the methodology proposed?

2) What is sophisticate about the search algorithm used in COPRA?

---

> ### Author Response · Authors · 2023-11-14
>
> Thank you for your valuable feedback. We respond to your comments and questions below:
>
> 1. “Why define the problem as an RL task, if this is not used in the methodology proposed?”
>
> **Ans.** Our approach is inspired by recent “verbal RL” approaches [1, 2] that use LLMs to perform control tasks (like game-playing) that were traditionally solved using RL. The big idea here is that unlike in traditional deep RL, there is no gradient-based policy update descent. Instead, RL-style exploration is combined with in-context learning: the agent interacts with the world, collects information, and then updates the LLM's prompt to elicit new behavior.
>
> The MDP formulation associates our approach with this perspective. The MDP captures **interactions** between our LLM and the external world (the proof environment), which form the primary difference between our approach and prior LLM-based approaches to theorem-proving. The sentence "Instead, we consider a fixed policy..." captures this essential difference between our method and classical RL approaches where one updates the policy iteratively.
>
> 2. “The structure of the framework proposed is not very clear. The algorithm in Fig. 3 is quite high-level. The calls and interactions with the LLMs are not discussed in much detail.”
>
> **Ans.** Figure 4 shows some of the interactions with the LLMs, we can add more details in the appendix. Please mention any specific detail you would like to see, and we would be happy to add that information to the paper.
>
> 3. "What is sophisticated about the search algorithm used in COPRA?”
>
> **Ans.** The search algorithm maintains a stack and keeps track of previous failures, and current execution feedback. This information is then added back to the prompt helping the guidance model (GPT-4) to better direct the search. The algorithm also uses a partial order over goals (originally introduced in Proverbot) to ensure that tactics actually simplify the proofs. Most of the previous approaches tend to use a stateless best-first search strategy, but our search uses a form of stack-based DFS and is stateful (by backtracking when the model starts repeating the same failed steps for any given proof state). This not only optimizes the number of inference steps needed to find a proof but also lets us fail fast when the proof is too hard to find (see Tables 1 and 2 for comparison).
>
> References:
>
> [1] Noah Shinn, Federico Cassano, Beck Labash, Ashwin Gopinath, Karthik Narasimhan, and Shunyu Yao. Reflexion: Language agents with verbal reinforcement learning. arXiv preprint arXiv:2303.11366, 2023.
>
> [2] Guanzhi Wang, Yuqi Xie, Yunfan Jiang, Ajay Mandlekar, Chaowei Xiao, Yuke Zhu, Linxi Fan, and Anima Anandkumar. Voyager: An open-ended embodied agent with large language models. arXiv preprint arXiv:2305.16291, 2023.

---

> > ### Comment · Reviewer_4t7K · 2023-11-22
> > **Follow up**
> >
> > Thanks to the authors for their replies and sorry for low responsiveness.
> >
> > - MDPs: these are described in Sec. 2.2 and mentioned only 2 other times in the paper, one in the conclusions. So, the point is: what do you do with these MDPs that appear (practically) nowhere else in the paper?
> >
> > - Algorithm: I think this should appear in the main body of the paper, rather than in the appendix, as it is one of the main contributions of the paper. But please correct me if I am wrong. I am afraid it is not up to me to decide what should appear in the algorithm or the level of detail.
> >
> > - Details: these are the details that it would be good to see in the description of the algorithm. I don't think these are mentioned anywhere in the paper right now, and therefore it is difficult to judge whether the algorithm is sophisticated or not.

---

> > > ### Author Response · Authors · 2023-11-22
> > > **Thank you! Here are the answers to your questions**
> > >
> > > We very much appreciate your response. Here are our responses to your points:
> > >
> > > *  MDP: The MDP defines the "search graph" that our algorithm systematically explores; without the MDP, there is no definition of what prover state (hypothesis + goal set) you transition to when you apply an LLM-selected tactic. The use of the MDP is therefore fundamental to our method. However, we understand that we can make this clearer in the paper. We can add a few sentences referring back to the MDP in Sections 3 and 4 of the paper.
> > >
> > > * Algorithm: We agree -- the algorithm (Figure 3) should definitely appear in the main body of the paper like it currently does. We were not proposing to move it but just offer additional details in the appendix. However, we agree that some of these details (such as how the use of the stack distinguishes us from prior methods) should appear in the main paper. We will make these changes in the revision.

---

### Official Review · Reviewer_VHEv · 2023-10-31

**Soundness:** 2 fair
**Presentation:** 3 good
**Contribution:** 3 good
**Rating:** 5
**Confidence:** 3

**Summary:**

This paper introduces COPRA, a language-agent framework that leverages the LLM GPT-4 for state-of-the-art performance in formal theorem-proving tasks. COPRA employs GPT-4 within a policy guiding a stateful backtracking search, where it selects proof tactics and retrieves relevant lemmas and definitions from an external database. The agent executes each tactic within a proof framework, using feedback to refine subsequent prompts, thus improving the decision-making process. Additionally, the system intelligently tracks search history to minimize errors and redundant queries to the language model. The experiments on two datasets verify the effectiveness of the proposed COPRA.

**Strengths:**

1. The paper is well organized with good language.
2. The addressed problem is interesting because it is a practical application of LLM.
3. The authors ensure the reproducibility of COPRA by providing detailed implementation details.

**Weaknesses:**

1. The method 'Decomposing the Enigma' [1], released in May 2023, appears to outperform COPRA on the miniF2F dataset, with a pass rate of 45.5% compared to COPRA's 23.36% [1]. More notably, 'Decomposing the Enigma' [1] achieves this using only ChatGPT-3.5, which raises questions about COPRA's claim to being 'state of the art.' Furthermore, COPRA's performance falls short when compared to Proverbot on the CompCert dataset.

2. It is unfair to compare the number of inferences made with REPROVER in Figure 5, as COPRA utilizes GPT-4 to prove theorems, while REPROVER employs a much smaller LLM. One query from GPT-4 is far more powerful than a single inference from REPROVER's LLM.

3. The authors seem to overstate their claim of having 'the first LLM-agent approach' with 'state-of-the-art' performance.

[1] Zhao, Xueliang, Wenda Li, and Lingpeng Kong. "Decomposing the Enigma: Subgoal-based Demonstration Learning for Formal Theorem Proving." arXiv preprint arXiv:2305.16366 (2023).

**Questions:**

1. Can you explain the performance gap mentioned in point 1 of the weaknesses?

2. Why does GPT-3.5 perform better than GPT-4 as indicated in Table 2? Does this suggest that there might be overfitting of prompts for different LLMs?

3. How can it be verified that theorems are proved sufficiently?

4. Why is COPRA claimed to be "the first LLM-agent approach to formal theorem-proving" when previous works like REPROVER and Decomposing the Enigma [1] might also be considered as LLM-agent approaches?

5. The reference page should begin on a new page (page 10).

6. What is the average API cost for COPRA per proven theorem?




[1] Zhao, Xueliang, Wenda Li, and Lingpeng Kong. "Decomposing the Enigma: Subgoal-based Demonstration Learning for Formal Theorem Proving." arXiv preprint arXiv:2305.16366 (2023).

---

> ### Author Response · Authors · 2023-11-14
> **Response (Part 1)**
>
> Thank you very much for providing valuable feedback. We answer your questions below. Due to the 5000 characters limit, we split our response into multiple comments.
>
> 1. Comparison with the method “Decomposing the Enigma” [1]. Comparison of COPRA's absolute performance with Proverbot on the CompCert dataset.
>
> **Ans.** “Decomposing the Enigma” [1] is based on Draft-Sketch-Proof (DSP) [4] style approaches, which uses informal proofs to find formal proofs.
> Having access to informal proofs (human-written or LLM-generated) simplifies the problem of synthesizing the formal proof into more of a translation problem, and that is one of the reasons why DSP-like approaches perform well on miniF2F datasets. It is important to note that these numbers are not comparable across different languages, [1] uses Isabelle which has powerful automatic reasoning tools like Sledgehammer, unlike Lean. As we mentioned in the related work section of our paper, this idea is orthogonal to the ideas in our paper, and we will explore ways to combine it with our methods in future work.
>
> That said, we also note that informal proofs are hard to get in real-world mathematical reasoning settings such as software verification like [CompCert](https://en.wikipedia.org/wiki/CompCert). Also, the approach prescribed in [1] starts with a manually written subgoal-based proof, and then subsequently refines it. By contrast, our approach is completely automatic.
>
> We agree that COPRA doesn’t outperform Proverbot on CompCert dataset in absolute numbers, but it uses much fewer inferences to find those proofs. It is quite possible that COPRA could do more proofs if we allowed more inference steps, but we are limited by our cost and API budget for GPT-4.
>
> 2. “It is unfair to compare the number of inferences made with REPROVER in Figure 5, as COPRA utilizes GPT-4 to prove theorems, while REPROVER employs a much smaller LLM. One query from GPT-4 is far more powerful than a single inference from REPROVER's LLM.”
>
> **Ans.** Our objective for comparing the number of inferences is as follows:
>
> a. We want to test the hypothesis of whether one can match the performance of fine-tuned models using just in-context learning. There is a recent trend of using Language Agent [2, 3] for performing control tasks using in-context “verbal RL”. We wanted to explore this idea in the context of theorem proving. It is important to note that ReProver is finetuned on proof-state and tactic pair data, but our approach is completely in-context.
>
> b. The intention behind pass@k-inferences is to assess the speed of the proposed method and the effectiveness of the LLM or neural network to guide the proof search. To avoid any confusion we plan to rename this metric as pass@k-guidance budget, which is a reasonable metric as it does the right trade-off in accounting for the time taken to complete a proof and at the same time ignores very low-level hardware details. Even if ReProver uses a smaller model, it can make a lot more inferences in a given time. However, we observe that ReProver is slower in finding proofs even with the wall clock time. This means that the effectiveness of the guidance model is important in quickly finding the proof. We think that Coq/Lean ITP run time contributes more significantly when the guidance model is not effective search-wise (i.e. predicts wrong tactics). We can present an argument for ReProver as to why it is slower. We know that on average the time taken per inference (which includes time to run on ITP as well) is around 1.55 seconds for ReProver and 6.73 seconds for Copra (From Table 2). Even if we assume ReProver doesn’t take any time (zero time) to run the inferences and spends all of 1.55 seconds running the tactic on ITP, then we can assume that about 5 seconds (6.73 - 1.55) is the average response time for GPT-4. However, we see that the number of inferences used by ReProver is about 46x higher on success. This interestingly shows up in the wall clock time too which is around 9x higher (~46x/5) for ReProver on success, so there is a tradeoff between the two, but the number of inferences dominates when the guidance model is not good. So, if the guidance model is good (it may be as big as GPT), we can empirically argue that asymptotically the search will converge to proof faster (given that it can be found using that guidance model).
>
> References:
>
> [1] Zhao, X., Li, W., & Kong, L. (2023). Decomposing the Enigma: Subgoal-based Demonstration Learning for Formal Theorem Proving. arXiv preprint arXiv:2305.16366.
>
> [2] Shinn, Noah, et al. "Reflexion: Language agents with verbal reinforcement learning." Thirty-seventh Conference on Neural Information Processing Systems. 2023.
>
> [3] Wang, Guanzhi, et al. "Voyager: An open-ended embodied agent with large language models." arXiv preprint arXiv:2305.16291 (2023).
>
> [4] Jiang, Albert Q., et al. "Draft, sketch, and prove: Guiding formal theorem provers with informal proofs." arXiv preprint arXiv:2210.12283 (2022).

---

> ### Author Response · Authors · 2023-11-14
> **Response (Part 2)**
>
> 3. “Why does GPT-3.5 perform better than GPT-4 as indicated in Table 2? Does this suggest that there might be overfitting of prompts for different LLMs?”
>
> **Ans.** GPT-3.5 doesn’t perform better than GPT-4. Table 2 compares the average wall clock time taken to find proofs using various approaches. What we observe is that GPT-3.5 response time is usually faster than GPT-4 per inferences which makes sense because it is presumably a smaller model than GPT-4. However, Copra agent with GPT-4 still finds proofs faster than GPT-3.5 because the search guidance provided by GPT-4 is more effective.
>
> 4. “How can it be verified that theorems are proved sufficiently?”
>
>
> **Ans.** COPRA generates formal proofs. Formal proofs are machine-checkable and Interactive Theorem Provers (ITP) like Lean are used for the purpose of verifying the correctness of [formal proofs](https://en.wikipedia.org/wiki/Formal_proof). Often formal proofs are much more rigorous and pedantic than informal proofs.
>
> 6. “Why is COPRA claimed to be "the first LLM-agent approach to formal theorem-proving" when previous works like REPROVER and Decomposing the Enigma [1] might also be considered as LLM-agent approaches?”
>
> **Ans.** Proof agents, like other Language Agents, need to have more human-like interactions with the interactive theorem prover (ITP) and generate proof step-by-step along with using external knowledge and tools. In approaches like ReProver, Baldur, PACT, etc., the model is not trained to interact with ITP. Instead, it is trained to either generate the whole proof all at once or one proof step at a time. Also, there is no use of execution feedback, history of proof so far, or known failed proof steps.
>
> Specifically, in the case of ReProver, there is no use of error feedback from ITP. Most of the current work uses a guidance model to carry out a search for completing the proof in a step-by-step manner. The guidance model is trained to predict the next step and the search is carried out using the best-first strategy. Some other methods generate the whole proof in one shot. Other approaches use informal proofs, for example in (Draft-Sketch-Proof) DSP [4].  While DSP might be using external knowledge (like generating informal proofs, hammer), there is no interaction with the ITP. The proof sketch is generated all at once and then other symbolic techniques (like hammers) are used to fill these holes. Also, in applications like software verification (e.g. our CompCert benchmark), the notion of informal proof is not well defined and there is no informal statement to begin with (which is required for approaches like DSP).
>
> Language Agents are used to perform control tasks (naturally framed using an MDP) through “verbal RL” [2, 3]. This is different from traditional deep RL because there is no gradient descent. Instead, the method synthesizes RL-style exploration with  in-context learning. Neither ReProver nor “Decomposing the Enigma” frame theorem-proving as a control/MDP task.
>
> 7. “The authors seem to overstate their claim of having  'state-of-the-art' performance.”
>
> **Ans.** Thank you for this point. We will make our claim of being state-of-the-art more nuanced, as we recognize that there are many versions of the state-of-the-art depending on how the problem is formulated (for example, whether the use of informal proofs is allowed or not).
>
> 8. “What is the average API cost for COPRA per proven theorem?”
>
> **Ans.** It depends on the number of inferences needed, the model used, and the length of the proof. It is somewhere from `$0.75-$5` per theorem on miniF2F and CompCert datasets.
>
> References:
>
> [1] Zhao, Xueliang, Wenda Li, and Lingpeng Kong. "Decomposing the Enigma: Subgoal-based Demonstration Learning for Formal Theorem Proving." arXiv preprint arXiv:2305.16366 (2023).
>
> [2] Noah Shinn, Federico Cassano, Beck Labash, Ashwin Gopinath, Karthik Narasimhan, and Shunyu Yao. Reflexion: Language agents with verbal reinforcement learning. arXiv preprint arXiv:2303.11366, 2023.
>
> [3] Guanzhi Wang, Yuqi Xie, Yunfan Jiang, Ajay Mandlekar, Chaowei Xiao, Yuke Zhu, Linxi Fan, and Anima Anandkumar. Voyager: An open-ended embodied agent with large language models. arXiv preprint arXiv:2305.16291, 2023.
>
> [4] Albert Q Jiang, Sean Welleck, Jin Peng Zhou, Wenda Li, Jiacheng Liu, Mateja Jamnik, Timothee´ Lacroix, Yuhuai Wu, and Guillaume Lample. Draft, sketch, and prove: Guiding formal theorem provers with informal proofs. arXiv preprint arXiv:2210.12283, 2022.

---

### Official Review · Reviewer_rua8 · 2023-10-31

**Soundness:** 2 fair
**Presentation:** 2 fair
**Contribution:** 2 fair
**Rating:** 5
**Confidence:** 3

**Summary:**

This paper introduces COPRA, a language-agent approach that prompts a large language model (LLM), specifically GPT-4, for formal theorem-proving. COPRA enhances the theorem-proving process by employing a stateful backtracking  policy search using language model. In particular, during the search, the policy selects proof tactics and retrieves essential information such as lemmas and definitions from an external database. Execution feedback and historical search data are then prompted again for policy update.  The authors tested COPRA on benchmarks like miniF2F and Coq tasks.

**Strengths:**

Formal theorem proving is a less explored application domain. This paper provides positive results for such an application.

**Weaknesses:**

1. Novelty. It seems that the method is similar to retrieval-based LLM in the sense that the policy uses an external database. It would be great if the authors could compare with this line of works in detail, especially the ReProver paper.

2. It seems that the proposed method does not significantly outperform ReProver.

3. Ablation studies might be needed to understand the role played by the RL component.

**Questions:**

1. Is there a difference between formal and informal theorem proving? It seems that there are some recent works that this work (LYRA: ORCHESTRATING DUAL CORRECTION IN AUTOMATED THEOREM PROVING) has a much higher score on miniF2F.

2. What is the role played by the RL part? I understand that you formulate the problem as an MDP and then the language-agent essentially mimics an RL algorithm. What is this particular RL algorithm? How to handle exploration-exploitation tradeoff?

3. Is RL really essential here? Can you replace it with other planning methods such as tree search, or even a close-loop planning method such as ReAct?

---

> ### Author Response · Authors · 2023-11-14
> **Response (Part 1)**
>
> Thank you for your valuable feedback. We address your main points below. Due to the 5000 characters limit, we split our response into multiple comments.
>
> 1. Questions about novelty and differences from existing retrieval-based LLM approaches like ReProver.
>
> **Ans.** Our approach is based on the new paradigm of Language Agents, where one uses LLMs to perform control tasks and interact with and external world. In particular, we were inspired by Reflexion [2], which uses language agents to solve natural language RL tasks, and Voyager [3], which uses language agents to play Minecraft. This approach is different from traditional deep RL because there is no gradient-based policy update. Instead, RL-style exploration is combined with in-context learning.
>
> In the proof synthesis setting, an LLM agent needs to have human-like interactions with the underlying interactive theorem prover (ITP) and generate a proof step-by-step along with using external knowledge and tools. In approaches like ReProver, Baldur, PACT, etc., the model is not trained to interact with the ITP. The proof steps are either generated all at once or one at a time, but there is no use of execution feedback, history of proof so far, or known failed proof steps.
>
> Specifically, in the case of ReProver, there is no use of error feedback from ITP. Most of the current work uses a guidance model to carry out a search for completing the proof in a step-by-step manner. The guidance model is trained to predict the next step and the search is carried out using the best-first strategy. Some other methods generate the whole proof in one shot. Other approaches use informal proofs, for example in Draft-Sketch-Proof (DSP) [4].  While DSP might be using external knowledge (like generating informal proofs, hammer), there is no interaction with the ITP. The proof sketch is generated all at once and then a powerful automatic reasoning tool (like hammer) is used to fill the holes. Also, it is important to note that in certain situations like software verification (e.g. CompCert) the notion of informal proof is not well defined and there is no informal statement to begin with (which is required for approaches like DSP).
>
> 2. “It seems that the proposed method does not significantly outperform ReProver.”
>
> **Ans.** It is true that the set of theorems that COPRA eventually discovers is not much larger than what Reprover discovers. However, COPRA  finds the proofs much faster than ReProver at least 9x faster in absolute wall clock time. COPRA also fails faster when it is not able to find the proofs. We weren't able to run COPRA on a very large number of timesteps because of the high cost of GPT-4 queries; however, we expect that doing so would improve COPRA's performance even more.
>
> More generally, we believe that as foundation models get more and more capable and cheaper (think of OpenAI's recent announcement regarding GPT-4-turbo, which significantly boosts GPT-4's context window while slashing costs), the benefits of COPRA-like approaches will become even clearer.
>
>
> 3. “Ablation studies might be needed to understand the role played by the RL component.”
>
> **Ans.** Please let us know any specific ablations that you would want to see, and we would be happy to run them.
>
> References:
>
> [1] Zheng, Chuanyang, Haiming Wang, Enze Xie, Zhengying Liu, Jiankai Sun, Huajian Xin, Jianhao Shen, Zhenguo Li, and Yu Li. "Lyra: Orchestrating Dual Correction in Automated Theorem Proving." arXiv preprint arXiv:2309.15806 (2023).
>
> [2] Noah Shinn, Federico Cassano, Beck Labash, Ashwin Gopinath, Karthik Narasimhan, and Shunyu Yao. Reflexion: Language agents with verbal reinforcement learning. arXiv preprint arXiv:2303.11366, 2023.
>
> [3] Guanzhi Wang, Yuqi Xie, Yunfan Jiang, Ajay Mandlekar, Chaowei Xiao, Yuke Zhu, Linxi Fan, and Anima Anandkumar. Voyager: An open-ended embodied agent with large language models. arXiv preprint arXiv:2305.16291, 2023.
>
> [4] Albert Q Jiang, Sean Welleck, Jin Peng Zhou, Wenda Li, Jiacheng Liu, Mateja Jamnik, Timothee´ Lacroix, Yuhuai Wu, and Guillaume Lample. Draft, sketch, and prove: Guiding formal theorem provers with informal proofs. arXiv preprint arXiv:2210.12283, 2022.

---

> > ### Author Response · Authors · 2023-11-14
> > **Response (Part 2)**
> >
> > 4. “Is there a difference between formal and informal theorem proving? It seems that there are some recent works that this work (LYRA: ORCHESTRATING DUAL CORRECTION IN AUTOMATED THEOREM PROVING) [1] has a much higher score on miniF2F.”
> >
> > **Ans.** A formal proof is something that can be automatically verified using an Interactive Theorem Prover (ITP), whereas an informal proof can only be verified by a human. ITP is a software tool to assist with the development of formal proofs by human-machine collaboration. This involves some sort of interactive proof editor, or other interfaces, with which a human can guide the search for proofs. Often formal proofs are much more rigorous and pedantic than informal proofs. So informal proof can be loosely considered as a proof sketch based on which one can write rigorous machine-checkable formal proofs.
> >
> > The LYRA [1] paper was announced after the ICLR deadline; however, we will be happy to cite this work in our final version. It uses the DSP [4] pipeline which uses informal proofs to find the formal proofs. This works very well on datasets like miniF2F which have problems from math competitions. However, real-world math formulations are not necessarily math competition problems with well-known informal proofs. Certain domains like software verification like [CompCert](https://en.wikipedia.org/wiki/CompCert) don’t have any notion of informal proofs. It is important to note that having access to informal proofs (human-written or LLM-generated) simplifies the problem of synthesizing the formal proof into more of a translation problem, and that is one of the reasons why DSP-like approaches perform well on miniF2F dataset. These approaches will work very well if the LLM can use its memory component to effectively recall the informal proof of a well-known math competition problem. However, a lot of formal theorem proving happens on a completely new set of formalizations which are heavily customized depending on the domain. For example, in the field of software verification, we will have a custom mathematical model defined for the software which has no informal proofs or definitions which can be recalled from memory by the LLMs. It is also important to note that the accuracy numbers in [1] are not directly comparable to ours because they use different proof languages. Isabelle is used in [1, 4] which has powerful automatic reasoning tools like Sledgehammer, unlike Lean.
> >
> > Our approach is generic and can work for any domain where formal theorem proving is used. It is one of the first methods which is domain and proof language agnostic.
> >
> > 5. “What is the role played by the RL part? I understand that you formulate the problem as an MDP and then the language-agent essentially mimics an RL algorithm. What is this particular RL algorithm? How to handle exploration-exploitation tradeoff?”
> >
> > **Ans.** Framing theorem proving as an MDP allows us to use the recent “verbal RL” techniques [2, 3] which use LLM for solving some natural language RL problems. This is different from traditional RL because there is no gradient descent, rather in-context Learning in LLM which resembles more like an exploration-only RL policy.
> >
> > 6. “Is RL really essential here? Can you replace it with other planning methods such as tree search, or even a close-loop planning method such as ReAct?”
> >
> > **Ans.** Learning from the environment (Interactive Theorem Prover) feedback is very important in fixing issues with the proof steps as one can see in the Figure 4 example. It is an interesting example that shows how keeping track of failures helps in fixing the error which cannot be achieved using a simple tree search. We can say that empirically, changing the prompt leads makes the policy explore different proof steps nudging the proof search in the right direction.
> > It is also important to note that knowing previously failed proof steps helps in nudging the LLM to look for alternatives that will work or pursue a different direction of search.
> >
> > References:
> >
> > [1] Zheng, Chuanyang, Haiming Wang, Enze Xie, Zhengying Liu, Jiankai Sun, Huajian Xin, Jianhao Shen, Zhenguo Li, and Yu Li. "Lyra: Orchestrating Dual Correction in Automated Theorem Proving." arXiv preprint arXiv:2309.15806 (2023).
> >
> > [2] Noah Shinn, Federico Cassano, Beck Labash, Ashwin Gopinath, Karthik Narasimhan, and Shunyu Yao. Reflexion: Language agents with verbal reinforcement learning. arXiv preprint arXiv:2303.11366, 2023.
> >
> > [3] Guanzhi Wang, Yuqi Xie, Yunfan Jiang, Ajay Mandlekar, Chaowei Xiao, Yuke Zhu, Linxi Fan, and Anima Anandkumar. Voyager: An open-ended embodied agent with large language models. arXiv preprint arXiv:2305.16291, 2023.
> >
> > [4] Albert Q Jiang, Sean Welleck, Jin Peng Zhou, Wenda Li, Jiacheng Liu, Mateja Jamnik, Timothee´ Lacroix, Yuhuai Wu, and Guillaume Lample. Draft, sketch, and prove: Guiding formal theorem provers with informal proofs. arXiv preprint arXiv:2210.12283, 2022.

---

> > > ### Comment · Reviewer_rua8 · 2023-11-22
> > >
> > > I would like to thank the authors for addressing my questions.
> > >
> > > **The role of RL** I am not convinced that RL is essential here. Of course, the problem can be written as an MDP -- the history is the state, the proof tactic is the action, reward is binary -- success or error. But this formulation does not touch upon the essence of RL -- in particular, it is unclear what kind of RL algorithm is implemented by in-context learning. This is different from existing works such as Tree of thoughts or RAP (Reasoning via Planning) which incorporated MCTS as part of the prompting strategy. In contrast, in this work, RL seems merely another way of saying the prompting is iterative -- the feedback is also added to the prompts.
> > >
> > > Thus, I would suggest conducting additional experiments to compare with those methods that incorporate tree search. The rebuttal period is limited. But it would be great if the authors could further clarify the role played in RL.
> > >
> > > **formal vs informal proof** I would like to thank the authors for explaining the difference between these two proofs. Suppose we have a dataset of informal proofs, can we incorporate them into the prompt and use in COPRA?

---

> > > > ### Author Response · Authors · 2023-11-22
> > > > **Thank you for your comment!**
> > > >
> > > > We appreciate your points. Our responses below:
> > > >
> > > > **RL**: The Reflexions paper (https://arxiv.org/abs/2303.11366) introduced the term "verbal RL" to describe what we are doing -- we were just following their usage. We believe that the use of the term "RL" is defensible here as RL, fundamentally, is about learning through exploration and rewards obtained for their actions. LLM-agent approaches like Reflexions and Copra are also exploring a world and receiving rewards. If you believe this is causing confusion, we can describe our approach in terms of LLM-based planning like in the RAP paper.
> > > >
> > > > A comparison with MCTS in the outer loop could be a good idea -- thank you for proposing this. We chose not to compare with other search algorithms as there is no prior work on LLM-guided search for formal theorem-proving. Also, at this point, only GPT-4 can reliably answer the sort of LLM queries Copra makes (as seen in our results), and repeated queries to GPT-4 are expensive. However,  a comparison with MCTS would certainly provide additional insights, and we will get working on that.
> > > >
> > > > **Formal vs. informal proofs**: Copra is fundamentally based on an underlying proof environment (Coq or Lean), which offers repeated feedback to the agent (in the form of updates to the goals/hypotheses) as the agent searches. As a result, its use is restricted to formal proofs. That said, it can take in additional natural language problem specifications or hints in addition to a formal proof goal -- these just go into the initial prompt.

---

### Official Review · Reviewer_VNR5 · 2023-11-03

**Soundness:** 2 fair
**Presentation:** 2 fair
**Contribution:** 3 good
**Rating:** 5
**Confidence:** 4

**Summary:**

The paper attacks the problem of tactic based theorem proving. Given a starting (goal,hypothesis) pair, we use *lean* to apply a *tactic* to it. This either yields a set of new goals, all of which need to be proved, or an error, or it directly proves the goal. The new set of goals are then handled recursively.

The paper proposes to choose the tactics by prompting an llm, and to embed this in a search algorithm which effectively does back tracking. The llm prompt includes *execution feedback* allowing the llm to improve on earlier failures.

The search is also guided by a way of pruning sets of (goal,hypothesis) pairs that are strictly harder to prove than existing ones.

More detailed comments:

Section 1

Some of the text in fig 1 is too small / blurry.

Sec 2.

The use of Sanchez-Stern's pruning method is cool.

It seems highly restrictive to only allow the model to choose tactics. Why not use prompting to ask the model the most promising partial prove to attack next?

The discussion of *Rewards* and *Histories and Policies* seems confusing and maybe erroneous. Detailed questions around that:
- Why are both scalar rewards and text feedbacks formalised as part of the reward ? How does this compare to traditional RL / MDP setups? Why the departure from that?
- Where is the scalar reward actually used? Apologies if I missed it but I couldn't quite see where the $r_i$ are used by the algorithm.
- What is the point of the sentence "A trajectory of a policy $\pi$ is a history ... ?

The use of execution feedback is nice, and it is nice to see this idea brought into theorem proving. The literature for program synthesis and other areas could be mentioned apropos this.

Seciton 3

Please fix the indentation of the pseudo-code in Fig. 3.

Fig. 4 was appreciated and seems helpful but it is slightly confusing in the current form ... Is this entire protocol repeated up to $k$ times, or is this all within one value of $j$ in your pseudo-code?

Section 4

End of page 5: including one shot prompting in copra dilutes what we can say about the method. Why not include an ablation where you only do the search, no one shot?

I'm not sure about removing ReProver's retrieval mechanism, could this not be done similarly to them? The code is available, and the agent system mentions a Lemma repository, which should be crucial to their model.

If you could integrate the lemma/retriever for MiniF2F the results should be even stronger, not sure why this couldn't be done, the explanation was unclear since you can access ReProver's code for MiniF2F and get the set of relevant premises from there (even just using BM25).

General comment: the results look fishy. I suspect GPT was trained on these datasets, and this would make all of the absolute comparisons with other methods hard to interpret. Please convince me otherwise. The passage on page 7 with paragraph heading "results" also strongly indicates this.

Pass @ k inferences is not intuitive to me:
-  why not use pass @ k tactic applications ? isn't this the main bottleneck? As it is structured, since each inference is restricted to a single response, they are essentially the same in this setup, but I think it's worth emphasising you are mostly restricted by the environment.
- doesn't this make the comparisons unfair also because GPT is more expensive than the other models (i.e. your Fig 6 x-axis is not comparable)? in the pass @ k inferences, why not use wall clock time then ?

Looking closely at Fig 6, why does proverbot have y value > 0 at x value = 0 ?

General comment: if we discount the absolute comparisons since GPT may have seen these datasets, how can we answer the research question "does the search strategy work". It seems like what might be missing is an answer to the questions "how does running Copra with k = 1 repeatedly with i.i.d. sampling of the LLM compare to copra with k > 1, normalised for number of tactic applications?".

In table 3:
- how does the "w/o backtracking" work, precisely?
- are these numbers comparable in terms of computational work? e.g. with number of tactic calls held constant?

Typos

- Page 7 typo (correlated [with the] number of correct proofs..)
- Typo in results (if only 60 inferences [are] allowed)

**Strengths:**

See the summary.

**Weaknesses:**

See the summary.

**Questions:**

Can you address the concerns? I like the paper and want to raise the score, but it feels like it might not be ready yet, if those concerns can't be reasonably addressed.

---

> ### Author Response · Authors · 2023-11-14
> **Response (Part 1)**
>
> Thank you for meticulously reviewing our paper and giving valuable feedback.  Due to the 5000 characters limit, we split our response into multiple comments. We respond to the important questions that you have raised below.
>
> 1. "It seems highly restrictive to only allow the model to choose tactics. Why not use prompting to ask the model the most promising partial proof to attack next?"
>
> **Ans.** If we understand your point correctly, you would like us to use the LLM to not just select a tactic for proving an obligation, but also for selecting the best obligation to prove. In most cases, the LLM has a large enough context window to see all obligations, this way there is an implicit choice available for the LLM to choose which obligation to attack first. In order to prove a theorem, all obligations have to be proved; in most cases, it might not matter which obligation is proved first (except for cases like induction where we prove the base case first).
>
> 2. Why are both scalar rewards and text feedback formalized as part of the reward? How does this compare to traditional RL / MDP setups? Why the departure from that? Where is the scalar reward actually used? What is the meaning of the sentence “A trajectory of policy …” in Section 2.
>
> **Ans.** The scalar reward is there to incentivize correct proofs and disincentivize incorrect proofs within the MDP framework. (In principle, the scalar reward function can also be used to incentivize shorter proofs.) We agree that the use of text feedback as part of the reward is not standard in traditional RL. The justification of this idea is that just like reward functions in traditional RL, the text describes the effectiveness of an action at a given state. We borrow this idea from recent work on “verbal RL” [1], which uses LLM agents for solving certain natural language RL problems and uses text as a generalization of rewards. This line of work departs from traditional RL in not having explicit policy updates, but combining RL-style exploration with in-context learning.
>
> We note, however, that the text feedback can also be modeled as part of the new environment state resulting from the action, as opposed to the reward signal. Such a model would make the reward function purely scalar just like in classical RL. If you think this is more natural, we would be happy to make this change.
>
> In RL, the agent's goal is to maximize the expected reward aggregated over trajectories. This requires a definition of trajectories. The paragraph  “A trajectory of policy ..,” offers this definition.
>
> 3. In the pseudo-code (Figure 3), is this entire protocol repeated up to k times, or is this all within one value of your j pseudo-code?
>
> **Ans.** Thank you for pointing this out, the ‘k’ mentioned in the algorithm is not the same ‘k’ as in pass@k-inference. All inference calls to LLMs even for fixing formatting errors are counted in pass@k-inferences. This is done to ensure that COPRA doesn’t have any unfair advantage over other approaches when we cap the number of inferences allowed. You can assume that the ‘k’ in the pseudo-code is just a free hyperparameter, and has nothing to do with the overall inference cap. The inference cap is strictly enforced. For pass@k-inference, COPRA can never make more than ‘k’ API calls to the LLM including all calls for fixing the formatting errors, incomplete responses, etc.
>
> 4. The use of one-shot prompting along with the agent interactions for finding proofs dilutes the capabilities of the agent.
>
> **Ans.**  Our reasoning here was that the goal is to find proofs quickly and cost-effectively. If GPT-4 can do it in one shot, the problem is not hard enough to justify the more expensive agent interaction.
>
>
> That said, if you like, we can run the Copra agent on 26 theorems which GPT-4 could do in one shot. Even in the most unlikely case of Copra not proving any of the 26 theorems, we can still say that COPRA does at least about 3% more proofs than what GPT-4 could do alone (which is evident from the numbers in Table 1).
>
> References:
>
> [1] Noah Shinn, Federico Cassano, Beck Labash, Ashwin Gopinath, Karthik Narasimhan, and Shunyu Yao. Reflexion: Language agents with verbal reinforcement learning. arXiv preprint arXiv:2303.11366, 2023.

---

> > ### Author Response · Authors · 2023-11-14
> > **Response (Part 2)**
> >
> > 5. Impact of removing retrieval from ReProver.
> >
> > **Ans.** At the time of writing the paper, the steps for running the retrieval augment version of ReProver were not publicly mentioned or published by the authors. There was no clarity on which index corpus was used for evaluating ReProver on the miniF2F dataset. However, we emailed the authors, and they shared the details steps and added it to their GitHub repository. We have now run ReProver with retrieval multiple times and got the results: 22.8, 24.0, 24.9, 24.9, 24.9%. This is less than their official numbers (26.5%), which can be because of subtle differences in hyperparameters (we have used the [method prescribed on their GitHub repo](https://github.com/lean-dojo/ReProver/blob/main/docs/eval_MiniF2F_ProofNet.md)). The difference between the retrieval-enabled ReProver and COPRA without retrieval is around 1%. More importantly, the number of inference steps increases on average (pass@k-inferences), which means retrieval doesn’t significantly improve the effectiveness of the search in ReProver even if it might have led ReProver to prove some more theorems. The average wall clock time taken is still 10x higher than that of COPRA. This is all when we discount ReProver extra inference calls to LLM for computing retrieval similarities and don’t include those inferences in pass@k-inferences.
> >
> >
> > If you like, we can compare ReProver with retrieval against COPRA with BM25 retrieval, and plot graphs with pass@k-inferences for the same. Note, however, that this may not be a fair comparison given the differences between the retrieval mechanism and the index corpus.
> >
> > 6. Data-leakage problem because of GPT-4 getting trained on the datasets.
> >
> > **Ans.** We note that this criticism could be applied to all generative AI approaches based on large pretrained models, whose pretraining data is rarely publicly accessible. For coding and language generation tasks, which have been studied in more depth, the use of large pretrained LLMs has now become standard, simply because the benefits of scale are simply too significant to ignore. We believe that AI-for-math is also taking a similar trajectory [3].
> >
> > Of course, this doesn't absolve us from the responsibility to thoroughly vet our results to make sure they don't just follow from train-test overlaps. We thought about this question carefully while working on this paper. Our conclusion was that data leakage isn't a significant contributor to our results, for several reasons.
> >
> > First, we note that Copra significantly outperforms one-shot invocations of GPT-4. If the results on Copra were significantly tainted by data leakage, we would have expected better performance from one-shot GPT-4.
> >
> > Second, not all the formal proofs of the miniF2F test dataset are available online. It is highly unlikely that GPT-4 has been trained on proof-state and tactic pair generated by hooking up the Lean Interactive Theorem Prover (ITP). Moreover, since the ground truth of miniF2F test for Lean is still not available, even if it were trained on proof-states one still needs to manually annotate ground truth tactics. Given that GPT-4 is a general-purpose LLM, it is highly unlikely that while training GPT-4 the miniF2F dataset was first manually annotated, and then proof-state and tactic pair information was collected by hooking up the Lean ITP.
> >
> > Also, in our agent interactions, we limit ourselves only to the goal at that point. There is no mention of the original theorem anywhere (except for the very first proof-state), so the chances that GPT-4 can correlate any intermediate state with the original theorem are very low unless it can somehow manage to simulate Lean’s interactive theorem proving within itself. It is also improbable that GPT-4 has seen the proof-state in the same format that we use, let alone using the execution feedback which has not been used in any known previous works.
> >
> >
> > One could hypothesize that some of the one-shot GPT-4 proofs might be influenced by potential training on the miniF2F dataset. However, this doesn’t seem to be true because we see that most of the proofs we generated were either not mentioned in the miniF2F test dataset or completely different from the manually written proofs in the miniF2F test dataset (including the first step mismatch). We can offer more details on these examples in the paper if you like.
> >
> > Finally, the ability of agent interactions to enhance the basic LLM approach seems to transcend OpenAI's LLMs. Subsequent to the submission of this paper, we ran COPRA on the recently released CodeLLama LLM. Copra improved CodeLlama’s capabilities to prove miniF2F’s theorem by 5%. This indicates that the in-context learning capabilities that we build are transferable and LLM-agnostic.
> >
> > References:
> >
> > [3] Azerbayev, Zhangir, et al. "Llemma: An open language model for mathematics." arXiv preprint arXiv:2310.10631 (2023).

---

> > > ### Author Response · Authors · 2023-11-15
> > > **Response (Part 3)**
> > >
> > > 7. The relevance of pass@k-inferences metric, and other alternatives like pass@k-tactics.
> > >
> > > **Ans.** We agree that pass@k-inferences can be misunderstood. The intention behind pass@k-inferences is to assess the speed of the proposed method and the effectiveness of the LLM or neural network to guide the proof search. To avoid any confusion we plan to rename this metric Pass@k-guidance-steps, which is a reasonable metric as it does a more even-handed trade-off in accounting for the time taken to complete a proof and at the same time ignores very low-level hardware details.
> > >
> > > It is also important to note that while GPT-4 model is much larger, individual calls are much more time-consuming and often subjected to long API throttling (evident from Table 2), unlike the locally running models. It is hard to estimate the computational cost, and smaller guidance models might be more time-efficient per inference. However, we think that Coq/Lean ITP run time contributes more significantly when the guidance model is not effective search-wise (i.e. predicts wrong tactics). We can present an argument for ReProver as to why it is slower. We know that on average the time taken per inference (which includes time to run on ITP as well) is around 1.55 seconds for ReProver and 6.73 seconds for Copra. Even if we assume ReProver doesn’t take any time (zero time) to run the inferences and spends all of 1.55 seconds running the tactic on ITP, then we can assume that about 5 seconds (6.73 - 1.55) is the average response time for GPT-4. However, we see that the number of inferences used by ReProver is about 46x higher on success. This interestingly shows up in the wall clock time too which is around 9x higher (~46x/5) for ReProver on success, so there is a tradeoff between the two, but the number of inferences dominates when the guidance model is not good. So, if the guidance model is good (it may be as big as GPT), we can empirically argue that asymptotically the search will converge to proof faster (given that it can be found using that guidance model). We will add a plot with a metric pass@k-seconds to show how pass@k-seconds correlates with pass@k-guidance. You can check the pass@k-seconds plot [here](https://ibb.co/gRvCbTc), from the [plot](https://ibb.co/gRvCbTc) (and Figure 5) it is clear that the two metrics are correlated.
> > >
> > > On the other hand, Pass@k-tactic is hard to implement. There is no easy way to restrict GPT-4 to generate only one tactic at a time. Even if we discard all other tactics generated, there is the possibility that we end up generating the same tactic again after searching for the next proof state because GPT-4 could already know that certain tactics only occur in certain patterns, and we end up wasting API calls. It is more inefficient timewise to artificially restrict the generated tactic. Compared to pass@k-tactic, pass@k-guidance is fairer because we need to account for bad formatting or incomplete responses that can be generated from the LLM. All the extra calls to fix any formatting error or incomplete responses are counted within the ‘k’ guidance call cap. It is also important to note that pass@k-tactic will be bad for longer proofs by definition.
> > >
> > >
> > > 8. Proverbot has y value > 0 at x value = 0 ?
> > >
> > > **Ans.** Thank you for pointing this out, it turns out that while calculating the number of inference steps for Proverbot9001, we marked guidance count for some theorems as N/A since when we failed to automatically download the proof-tree from Proverbot9001 results page ([Proverbot Report (ucsd.edu)](https://proverbot9001.ucsd.edu/reports/2019-11-20T18d38d02-0700+cd762eb9e7e6e44153bd766654727a36a3dcad0b/report.html)). This happened for 7 theorems, we will manually download the proof tree and parse it to find out the number of guidance steps. This will only increase the average steps needed by Proverbot to finish proofs.
> > >
> > > References:
> > >
> > > [1] Noah Shinn, Federico Cassano, Beck Labash, Ashwin Gopinath, Karthik Narasimhan, and Shunyu Yao. Reflexion: Language agents with verbal reinforcement learning. arXiv preprint arXiv:2303.11366, 2023.
> > >
> > > [2] Guanzhi Wang, Yuqi Xie, Yunfan Jiang, Ajay Mandlekar, Chaowei Xiao, Yuke Zhu, Linxi Fan, and Anima Anandkumar. Voyager: An open-ended embodied agent with large language models. arXiv preprint arXiv:2305.16291, 2023.
> > >
> > > [3] Azerbayev, Zhangir, Hailey Schoelkopf, Keiran Paster, Marco Dos Santos, Stephen McAleer, Albert Q. Jiang, Jia Deng, Stella Biderman, and Sean Welleck. "Llemma: An open language model for mathematics." arXiv preprint arXiv:2310.10631 (2023).

---

> > > > ### Author Response · Authors · 2023-11-15
> > > > **Response (Part 4)**
> > > >
> > > > 9. How does running Copra with k = 1 repeatedly with i.i.d. sampling of the LLM compare to copra with k > 1, normalised for number of tactic applications?
> > > >
> > > > **Ans.** Since the temperature is set to zero at all times whether we do it for the agent or for one-shot. k > 1 sample will not significantly change anything. Temperature zero also ensures reproducibility, highly deterministic, and focused responses for a given prompt as per [OpenAI documentation](https://platform.openai.com/docs/api-reference). Also, parts from answer 3 apply to this question as well because we found completely new proofs for the theorems that have their proofs mentioned in the miniF2F repository. We also saw that the COPRA approach worked for other LLMs like CodeLlama and enhanced its capability to prove more theorems.
> > > >
> > > > 10. How does “w/o backtrack” ablation work? Comparison of the amount of computation in Table 3.
> > > >
> > > > **Ans.** Without backtracking works by failing as soon as all possibilities of the next possible proof step at any given proof state are exhausted i.e.LLM starts to repeat the same failed proof again. Since “backtracking” will always end up trying more proof steps than “w/o backtracking” the computation done at search time will be more with backtracking. However, it is possible that the final proof found using backtracking has less number of tactics than incomplete proof found using “w/o backtracking”.
> > > >
> > > >
> > > > We also appreciate your efforts in pointing out typos and some minor corrections in the paper. We will fix those issues in the revision.
> > > >
> > > > References:
> > > >
> > > > [1] Noah Shinn, Federico Cassano, Beck Labash, Ashwin Gopinath, Karthik Narasimhan, and Shunyu Yao. Reflexion: Language agents with verbal reinforcement learning. arXiv preprint arXiv:2303.11366, 2023.
> > > >
> > > > [2] Guanzhi Wang, Yuqi Xie, Yunfan Jiang, Ajay Mandlekar, Chaowei Xiao, Yuke Zhu, Linxi Fan, and Anima Anandkumar. Voyager: An open-ended embodied agent with large language models. arXiv preprint arXiv:2305.16291, 2023.
> > > >
> > > > [3] Azerbayev, Zhangir, Hailey Schoelkopf, Keiran Paster, Marco Dos Santos, Stephen McAleer, Albert Q. Jiang, Jia Deng, Stella Biderman, and Sean Welleck. "Llemma: An open language model for mathematics." arXiv preprint arXiv:2310.10631 (2023).

---

> ### Author Response · Authors · 2023-11-21
> **Results from the additional experiments**
>
> Dear Reviewer,
>
> We ran experiments to address the concerns raised by you. We are excited to share some findings with you, which we believe demonstrate significant progress:
> 1. **Enhancement in Problem Solving Capability using Retriever**: In response to the retriever disabled issue, we developed a BM25 retriever and re-ran our experiments. This gives us **27.45%** i.e. 67/244 success rate on miniF2F. This outperforms notable systems like ReProver, Llemma, PACT, and Lean GPT-f.
>
> 2. **Addressing Data Leakage Concerns**: We undertook a thorough analysis of potential data leakage within the miniF2F dataset. Our findings indicate that of the 67 proofs our system generated, 68.65% are distinct from those mentioned in the miniF2F dataset or are cases where the dataset lacks a corresponding proof. Notably, the proofs matching with miniF2F are primarily simple ones, solvable with a single tactic like 'linarith', 'norm_num', or 'nlinarith'. Excluding these straightforward cases, we found that 92% of our generated proofs are unique compared to the miniF2F dataset.  Additionally, 25.37% of the proofs generated by our approach are not mentioned in the miniF2F dataset, compared to 22.9% of for the ReProver.
>
> We appreciate the opportunity to run additional experiments and hopefully address your concerns regarding retriever and data leakage.
>
> Best regards,
>
> Authors

---

### Author Response · Authors · 2023-11-20

Dear Reviewers,

Thank you for your valuable insights and suggestions. We have tried our best to answer your questions in our author response, and we are also working to revise the paper following your suggestions. Given that the discussion period is ending soon, we were wondering if you could let us know if you have further questions or whether the authors' response addressed your concerns. We would be delighted to answer any further questions that you might have.

Best regards,

Authors

---

### Author Response · Authors · 2023-11-23
**Summary of changes in the revision**

We changed the following in our rebuttal revision:
1. Changed the MDP formulation to a traditional formulation with scalar rewards.
2. Changed the metric name from `pass@k-inferences` to `pass@k-guidance-steps`.
3. Added results from additional experiments:

      3.1. Experiments with retrieval enabled Copra and ReProver on the miniF2F dataset.

      3.2. Experiments with CodeLlama as an underlying model for Copra, to show the transferability of our approach.

      3.3  Experiments with running Copra on theorems which are successfully proved by one-shot baselines.
4. Additional analysis for:

      4.1 Extensive analysis of the generated proofs for data Leakage problem due to potential training of GPT-4 on miniF2F.
5. Minor changes:

    5.1 Fixed the typos.

    5.2 Fixed blurry text in images.

    5.3. Added more clarification on comparison with methods that use informal proofs.

---

### Meta-Review · Area_Chair_gHSo · 2023-12-15

**Metareview:**

This paper considers theorem proving as a reinforcement learning problem here a proof is viewed as a sequence of actions that either close the proof (succeed) or fail.  The standard metric of pass@k used to evaluate automated provers os modified to pass@q where q is the number of calls to an LLM to propose a next action.

This paper got four ratings of 5.  I expect I would have rated it higher on the grounds that the use of LLMs. A main complaint of the reviewers is that the performance seems to be worse or equivalent to other systems.  The paper is using a new metric of pass@q which makes comparisons hard.  But they should have made an attempt to evaluate pass@k for comparison purposes.

**Justification For Why Not Higher Score:**

The empirical results are difficult to interpret and therefore not compelling.

**Justification For Why Not Lower Score:**

This is the lowest score.

---

### Decision · Program_Chairs · 2024-01-16

Reject